# ON DISCRIMINATIVE VS. GENERATIVE CLASSIFIERS: RETHINKING MLLMS FOR ACTION UNDERSTANDING

**Zhanzhong Pang**[1], **Dibyadip Chatterjee**[1], **Fadime Sener**, **Angela Yao**[1]
[1]National University of Singapore
{pang, dibyadip, ayao}@comp.nus.edu.sg, famesener@gmail.com

## ABSTRACT

Multimodal Large Language Models (MLLMs) have advanced open-world action understanding and can be adapted as generative classifiers for closed-set settings by autoregressively generating action labels as text. However, this approach is inefficient, and shared subwords across action labels introduce semantic overlap, leading to ambiguity in generation. In contrast, discriminative classifiers learn task-specific representations with clear decision boundaries, enabling efficient one-step classification without autoregressive decoding. We first compare generative and discriminative classifiers with MLLMs for closed-set action understanding, revealing the superior accuracy and efficiency of the latter. To bridge the performance gap, we design strategies that elevate generative classifiers toward performance comparable with discriminative ones. Furthermore, we show that generative modeling can complement discriminative classifiers, leading to better performance while preserving efficiency. To this end, we propose **G**eneration-**A**ssisted **D**iscriminative (GAD) classifier [1] for closed-set action understanding. GAD operates only during fine-tuning, preserving full compatibility with MLLM pretraining. Extensive experiments on temporal action understanding benchmarks demonstrate that GAD improves both accuracy and efficiency over generative methods, achieving state-of-the-art results on four tasks across five datasets, including an average 2.5% accuracy gain and $3\times$ faster inference on our largest COIN benchmark.

## 1 INTRODUCTION

Video understanding has traditionally focused on closed-set recognition and detection (Kay et al., 2017; Damen et al., 2022). Recent advances in Multimodal Large Language Models (MLLMs) (Li et al., 2023; Maaz et al., 2023; Wang et al., 2024) have expanded the scope to open-world settings, enabling free-form language output via autoregressive (AR) token generation. This language-centric design enables MLLMs to solve diverse video tasks through text output, providing a general and task-agnostic framework. This motivates examining their utility for conventional classification, where textual outputs can enrich semantics while avoiding task-specific architectures.

To extend MLLMs to classification tasks in video understanding, prior works (Hu et al., 2023; He et al., 2024; Chen et al., 2024a; Wu et al., 2024; Chatterjee et al., 2025) cast these tasks as generative problems, employing MLLMs as *Generative Classifiers* (Jaini et al., 2023). In this formulation, models are prompted with queries such as "What is the action in the video?" and fine-tuned to autoregressively generate concise action labels (e.g., "add onion") as free-form text. Yet the generative objective of MLLMs is not inherently tailored for classification, and the effectiveness of this reframing remains underexplored. In contrast, *Discriminative Classifiers* (Ng & Jordan, 2001) align more naturally with classification, learning task-specific representations and predicting actions directly, thus avoiding the additional step required to map free-form text back to predefined classes (Chen et al., 2024a). Exploring discriminative learning with MLLMs for video understanding therefore remains a promising yet under-investigated direction.

In this paper, we demonstrate the advantages of discriminative classifiers over generative ones for temporal action understanding. We adapt pre-trained MLLMs into discriminative classifiers by appending a learnable token to the visual input, enabling the model to encode a global representation

---

[1]Code: https://github.com/pangzhan27/GAD.

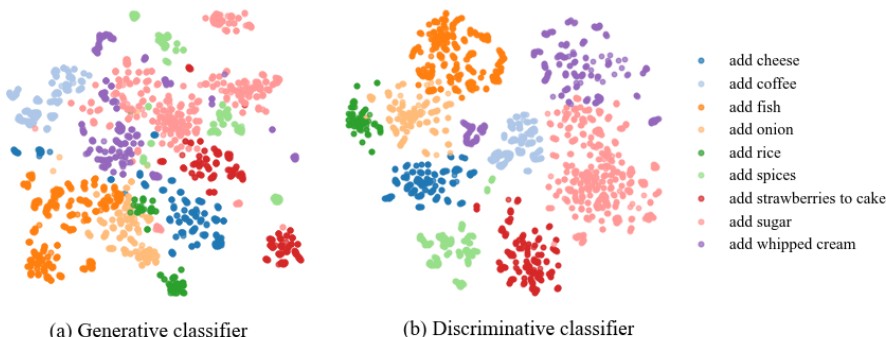

(a) Generative classifier  (b) Discriminative classifier

Figure 1: T-SNE plot comparing feature spaces of generative and discriminative classifiers on CrossTask dataset for actions sharing the verb 'add'. Generative feature - mean of output token features; Discriminative feature - the learnable token feature.

for classification. Our discriminative approach surpasses generative classifiers (Lin & Shou, 2025; Chen et al., 2024a) in both accuracy and efficiency. By directly optimizing decision boundaries (Fig. 1), the discriminative formulation reduces action confusion, leading to consistent performance gains. Furthermore, it predicts an action (*e.g.* "add strawberries to cake") in a single forward pass, compared to the multiple passes required by generative classifiers to autoregressively output tokens (*e.g.* {'add', ' strawberries', ' to', ' cake'}).

The superior performance of the discriminative classifier motivates an investigation into why it outperforms the generative one. Generative classifiers are more prone to confuse semantically similar actions, such as "add onion" and "add rice". This stems from the fact that action labels are intentionally annotated to be concise and high-level rather than descriptive (Zhou et al., 2024; Kay et al., 2017), which produces substantial semantic overlap across classes, *e.g.* the frequent use of verbs like "add" and "put". As a result, beyond the inherent visual ambiguities in video input, semantic overlap in the output space poses an additional challenge for generative classifiers. In contrast, the discriminative formulation ignores label semantics (Ye & Guo, 2017), eliminating semantic overlap and achieving clearer separation between actions. We further show that the generative and discriminative classifiers can become functionally equivalent when the action labels are introduced into the tokenizer's vocabulary as singular tokens, and decoded in one autoregressive step. Adding single tokens prevent action labels being tokenized into subwords shared across actions, thus removing semantic overlap. This finding highlights the potential of extending MLLMs as generative classifiers to learn task-specific representations for discriminative learning.

However, the task-specific discriminative classifiers lose the semantic richness conveyed by generated text. This motivates us to incorporate generative modeling to complement and thus enhance discriminative learning. To this end, we propose a **G**eneration-**A**ssisted **D**iscriminative (GAD) classifier for temporal action understanding, which integrates discriminative objectives with auxiliary generative objectives within a single end-to-end framework. This design maintains the strengths and efficiency of discriminative learning while incorporating additional semantics and contextual information through generative modeling, enabling context-aware and semantically enriched representations.

We investigate a broad spectrum of temporal action understanding tasks, spanning basic step and task recognition and step forecasting (Tang et al., 2019) to the more challenging setting of online action detection (Zhukov et al., 2019; Damen et al., 2022; Song et al., 2023). Our experiments demonstrate that the discriminative classifier achieves higher accuracy and lower inference latency than the generative one. Within the proposed GAD framework, generative modeling further strengthens classification by regularizing training with semantic encoding and contextual enrichment, while inference relies solely on classification, preserving the efficiency of discriminative learning. Notably GAD achieves state-of-the-art across on four tasks across five datasets, including an average 6.8% F1 gain and 1.8x speedup on EPIC-Kitchens-100, 1.5% F1 gain and 3x speedup on Ego4D GoalStep, and 2.5% Top-1 accuracy gain with 3x speedup on COIN.

Overall, our contributions are summarized as:

- We demonstrate that generative classifiers underperform discriminative ones on classification tasks, primarily due to semantic overlap in the generative (textual) output space.

- We align generative and discriminative classifiers by interpreting classification as a single-step generation process, where predefined action labels are introduced as new tokens.
- We propose a generation-assisted discriminative (GAD) framework, showing that auxiliary generative objectives enrich discriminative learning and improve temporal action understanding.

## 2 RELATED WORKS

**Temporal action understanding** involves recognition tasks such as temporal action detection and segmentation (Zhao et al., 2017; Ding et al., 2023), step recognition and forecasting (Tang et al., 2019), and video recognition (Zhang et al., 2021). Classical approaches model temporal dynamics using temporal convolutional networks (Farha & Gall, 2019), recurrent neural networks (Xu et al., 2019), or transformers (Liu et al., 2022). Recently, MLLMs have shown strong performance by formulating recognition as an autoregressive token generation task, where labels are expressed as free-form text and decomposed into tokens (Hu et al., 2023; He et al., 2024). Chen et al. (2024a); Wu et al. (2024); Chatterjee et al. (2025) extended MLLMs to real-time interaction, temporal summarization, and forecasting, while Ye et al. (2025) applied a QA-style generation approach for video action recognition. However, autoregressive token generation is inefficient, and its effectiveness for recognition tasks remains underexplored. Our work demonstrates that generative learning underperforms its discriminative counterparts due to the added semantic complexity in text output.

**Customized tokenization** may use task-specific or optimized strategies (Liu et al., 2024; Zhang et al., 2025) to reduce token length for faster decoding and improved semantic representation. Using Ego4D videos (Grauman et al., 2022), Lin & Shou (2025) constructed a hierarchical vocabulary of video narrations to enable faster inference. In image retrieval, Caron et al. (2024); Zhang et al. (2024) designed language-based discriminative entity codes, converting images into compact and semantically rich tokens serving as identifiers for efficient and accurate retrieval. For multiple-choice question answering, mapping each answer option to a single symbol effectively tokenizes answers into one token, enabling faster inference (Joshua Robinson, 2023; Ranasinghe et al., 2024). However, in action understanding, fine-grained actions that share verbs and objects make distinctive tokenization difficult, and preserving full semantics is harmful. Similar observations have been made in closed-set image classification (Cooper et al., 2025; Conti et al., 2025), where distinguishing fine-grained categories remains challenging for LLMs. Specialized prompting has been proposed to improve differentiation, but the approach remains generative. In contrast, we adopt a classification-style approach, equivalent to encoding each action as a unique, unstructured atomic code.

**Unified retrieval and generation** within a single model represents a key step toward building general-purpose multi-task systems. Earlier vision-language foundation models, (Yu et al., 2022; Li et al., 2022; Chen et al., 2024b), constructed hybrid frameworks that combine a vision-text encoder and a text decoder, pretrained jointly with contrastive and language modeling objectives. Large language models (LLMs) (Koh et al., 2023; Ma et al., 2024) have also shown potential as unified backbones when trained jointly with both losses, which can be further enhanced with task-specific instruction tuning (Muennighoff et al., 2024). Our work investigates MLLMs for classification tasks, aligning discriminative and generative learning within the same task, in contrast to existing approaches that address them separately for different tasks.

## 3 METHOD

We present the Generation-Assisted Discriminative (GAD) classifier, which leverages pre-trained LLMs for fine-tuning on downstream temporal action understanding tasks. Our focus is on the fine-tuning stage because general MLLMs typically struggle with task-specific requirements and must be further adapted for optimal performance.[2] We first formalize the problem of temporal action understanding, then introduce and analyze the generative classifier baseline and its challenges, before motivating our proposed solutions.

### 3.1 PRELIMINARIES

We focus on fine-tuning pre-trained LLMs for temporal action understanding in a closed-set classification setting. Given a video sequence $\mathcal{V}$, which may be a short clip or an entire video sampled at a

---

[2]Qwen2.5-VL-7B only achieves 16.1% and 8.9% zero-shot accuracy on COIN step and next-action prediction, respectively, compared to 67.3 % and 51.6% after fine-tuning. Find more results in Appendix C.

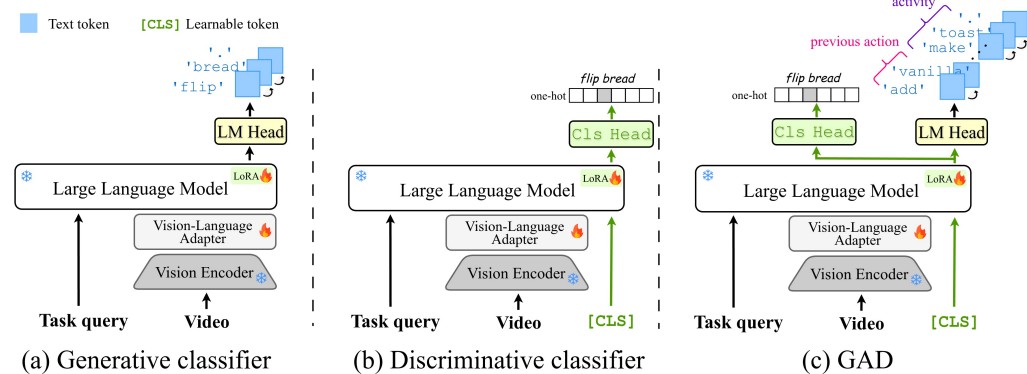

Figure 2: Comparison between different architectures for downstream video-related recognition tasks: (1) Generative classifier: treating action labels as free text. (2) Discriminative classifier: learning an extra representation for downstream tasks. (3) Generation-assisted discriminative (GAD) classifier: learning an extra representation that is regularized through task-related generation.

predefined rate, and a task query $\mathcal{Q}$, the objective is to predict a label $\boldsymbol{y}$ from a predefined category set $Y$. This problem formulation is general, accommodating a wide range of recognition scenarios.

**Action labels.** Temporal action understanding tasks typically involve actions that unfold over time and exhibit temporal correlations. These actions are generally expressed as concise and simplified verb-noun phrases (*e.g.* "add sugar"), and sometimes enriched with additional contextual elements (*e.g.* "add strawberries to cake"), to capture both the action and objects involved. The action vocabulary is typically closed, consisting of recurring motions ("add", "screw") applied to a shared set of objects, which naturally leads to semantic overlap across labels.

## 3.2 ARCHITECTURE

We adopt a LLaVA-style (Liu et al., 2023) architecture as the backbone, comprising a visual encoder $E_v$, a language decoder $D_t$, and a vision-language adaptor $A_{vt}$. Video frames $\mathcal{V}$ are first encoded by the vision encoder $E_v$, then mapped to text-aligned tokens via the vision-language adaptor $A_{vt}$, yielding visual tokens $\mathcal{F}_v = A_{vt}(E_v(\mathcal{V}))$. Meanwhile, the task query $\mathcal{Q}$ is tokenized into text tokens $\mathcal{F}_t$, enabling joint processing with the visual tokens by the language decoder. During fine-tuning, only the vision–language adaptor and the language decoder are trained using LoRA, while all other components remain frozen.

**Generative Classifier Baseline.** In a generative classifier, text and visual tokens are concatenated and fed into the causal language decoder to autoregressively generate subword tokens using a language modeling head. During training, the generative classifier is provided with a target action label $y$, which is tokenized by the language model's tokenizer into a sequence of subwords $\{u_0, u_1, \cdots, u_{n-1}\}$. Here, $n$ represents the total number of subword tokens representing $y$, and these subwords are drawn from the tokenizer's vocabulary. For example, the action label "take pancake from pan" is tokenized as $\{$ *'take', ' panc', 'ake', ' from', ' pan'* $\}$ by the Llama-3 tokenizer (Grattafiori et al., 2024). The model is trained to maximize the conditional probability of each target subword $u_i$ over the vocabulary, conditioned on the visual and textual inputs and previously generated subwords $u_{<i}$. The training is done using the standard language modeling loss $\mathcal{L}_{gen}$, *i.e.* the negative log-likelihood, applied to each subword token.

$$u_i = g_\omega(\mathcal{Q}, \mathcal{V}, u_{<i}) = D_t(\mathcal{F}_t \oplus \mathcal{F}_v \oplus u_{<i}), \quad \mathcal{L}_{gen} = -\sum_i \log P_r(u_i \mid \mathcal{Q}, \mathcal{V}, u_{<i}, \omega) \quad (1)$$

where $g_\omega$ is the generative classifier parameterized by $\omega$, $\oplus$ denotes concatenation. When applying the generative classifier for closed-set classification, the decoder is finetuned to generate subword tokens from a fixed vocabulary subset that covers the subwords required to represent all action labels.

**Discriminative Classifier.** Here, we extend the generative model to a discriminative model for downstream classification tasks. Instead of directly classifying video features using specialized models, we repurpose the existing MLLM as a task-agnostic classifier, eliminating the need for task-specific architectures. A learnable `[CLS]` token (Ye et al., 2025; Lin & Shou, 2025) is appended to the end of the language model input sequence, attending to all preceding tokens and generating

representations that integrates both the video input and the task query. We find that using a learnable token enhances generalization, whereas relying on visual tokens directly can lead to overfitting (see Appendix B.4). The resulting output representation $o$ is fed to customized classification heads for the downstream classification tasks by optimizing the traditional cross-entropy loss. Note that the language modeling head is disabled in this case, since generating subwords is unnecessary.

$$o = f_\phi(\mathcal{Q}, \mathcal{V}, \texttt{[CLS]}) = \text{D}_\text{t}(\mathcal{F}_\text{t} \oplus \mathcal{F}_\text{v} \oplus \texttt{[CLS]}), \quad \mathcal{L}_\text{cls} = -\log \text{P}_\text{r}(\text{y} \mid \text{o}, \phi'), \qquad (2)$$

where $f_\phi$ denotes the discriminative classifier parameterized by $\phi$, and $\phi'$ represents the parameters of the classification head.

This discriminative formulation supports faster inference in low-latency scenarios. It also prevents action labels from being tokenized into subwords, avoiding semantic overlap from shared subwords across labels. In Sec. 4, we show that such overlap causes discriminative and generative classifiers to behave differently, with the generative one more prone to confusing verbally similar actions. Removing this overlap through specific designs, such as isolated tokenization, reduce these confusions among similar actions. In fact, when action labels are excluded from the text query and only used as output target, the discriminative classifier becomes a special case of a generative classifier by 1) adding action labels as new vocabulary entries. 2) merging the classification head into the language modeling head. Since these added entries are not used as input, their embeddings can be initialized randomly without affecting performance, and classification is equivalent to generating the entire action label in one single step. This equivalence demonstrates the potential of extending existing MLLMs to learn task-specific representations for discriminative learning.

**Generation-Assisted Discriminative Classifier.** Generative modeling still remains valuable by providing deeper semantic cues through text generation, effectively encoding label semantics and context. To combine these benefits with the discriminative classifier, we propose the Generation-Assisted Discriminative (GAD) classifier, a unified framework that augments the the discriminative model by adding the language modeling head, to produce auxiliary generative outputs.

We investigate three strategies for unification: (1) sequential learning with discriminative learning first, followed by generation conditioned on the learned representation. (2) sequential learning with generation first, followed by representation learning conditioned on the generated output. (3) parallel learning, where both are learned simultaneously using shared text and visual input. We focus on the first strategy here as it proves most effective, leaving details of other strategies to Appendix B.2. Specifically, for (1), discriminative learning follows the standard discriminative classifier setup, after which generative modeling is applied conditioned on the video $\mathcal{V}$, the task query $\mathcal{Q}$, and the learnable token $\texttt{[CLS]}$. The generation loss in Eq. 1 is then adapted as

$$\mathcal{L}'_{gen} = -\sum_i \log P_r(u_i \mid u_{<i}, \mathcal{Q}, \mathcal{V}, \texttt{[CLS]}, \theta), \qquad (3)$$

where $\theta$ denotes the learnable parameters in this model, and $u_i$ represents the $i^{th}$ tokenized subword corresponding to the generation target. The overall training loss combines the adapted generation loss and the classification loss from Eq. 2, weighted by a balance factor $\lambda$, as $\mathcal{L}_{GAD} = \mathcal{L}_{cls} + \lambda\mathcal{L}'_{gen}$.

We use generative modeling as an auxiliary task to regularize the representation learning in the discriminative classifier. For instance, it can implicitly capture intentions or reason about past and future steps, thereby supporting recognition of the current action in goal-oriented procedural videos. While this formulation could also treat discriminative and generative learning as separate, independent tasks, prior work shows that simply unifying their objectives in a single model offers limited benefit (Ma et al., 2024; Muennighoff et al., 2024). In contrast, our auxiliary-task setting leverages generative modeling specifically to enhance discriminative learning.

### 3.3 Model Training & Inference

During training, we leverage pre-trained vision encoders and LLMs and perform instruction tuning with task-specific queries. The vision encoder is frozen; the vision-language adapter is fine-tuned, and the LLM is updated via LoRA (Hu et al., 2022). We optimize both discriminative and generative objectives across all training instances, with the latter serving as auxiliary supervision to enrich representation learning. At inference, we disable the generative branch and use only the discriminative classifier to produce final predictions.

Table 1: Generative (Gen) vs. discriminative (Disc) classifier on OAD tasks. The runtime analysis is performed on a single NVIDIA RTX A5000 GPU.

| LLM | Model | THUMOS'14 | | CrossTask | | EPIC-Kitchens-100 | | Ego4DGoalStep | |
|-----|-------|-----------|-----|-----------|-----|-------------------|-----|---------------|-----|
| | | S-F1 / P-F1 | FPS | S-F1 / P-F1 | FPS | S-F1 / P-F1 | FPS | S-F1 / P-F1 | FPS |
| Llama3.2- | Gen | 56.9 / 38.8 | 38.3 | 46.8 / 31.7 | 44.0 | 16.7 / 13.9 | 28.8 | 8.9 / 3.4 | 17.8 |
| 1B-Instruct | Disc | **57.8 / 40.1** | **58.0** | **48.8 / 34.0** | **59.4** | **23.2 / 19.3** | **51.1** | **10.6 / 4.1** | **53.6** |
| Qwen2.5- | Gen | 55.8 / 38.9 | 30.2 | 44.0 / 28.6 | 36.3 | 16.2 / 13.4 | 26.6 | 8.8 / 3.2 | 12.3 |
| 0.5B-Instruct | Disc | **57.3 / 39.6** | **48.8** | **45.3 / 29.6** | **52.8** | **22.0 / 18.3** | **48.6** | **9.9 / 3.4** | **52.1** |

## 4 EXPERIMENTS

Our experiments aim to answer the following research questions for discriminative and generative classifiers: (1) How do they compare in performance? (2) What factors influence their performance? (3) When and how can a generative classifier enhance a discriminative one?

### 4.1 DATASETS, EVALUATION, AND IMPLEMENTATION DETAILS

**Datasets & Tasks:** We evaluate on four temporal action understanding tasks, including step recognition, step forecasting, task (activity) recognition, and the more challenging setting of online action detection across five datasets. **Step recognition** identifies the occurred action in a given video clip, while **Step forecasting** anticipates the upcoming action in the clip. The COIN dataset (Tang et al., 2019) is adopted for these tasks, as it contains fine-grained, goal-oriented action steps. **Task Recognition** aims to detect the overall activity category of a given video with multiple steps. COIN also supports this task, providing high-level labels that reflect the hierarchical structure of such activities. **Online Action Detection (OAD)** focuses on recognizing actions in a streaming video using only past observations. Wideely used OAD datasets, including THUMOS'14 (Idrees et al., 2017), EPIC-Kitchens-100 (Damen et al., 2022), and CrossTask (Zhukov et al., 2019), cover both sport videos with loosely related actions as well as procedural videos with more correlated and fine-grained actions. We also include Ego4D GoalStep (Song et al., 2023), which features more descriptive labels than the concise ones in the other datasets, allowing us to assess the impact of label complexity on model performance.

**Evaluation.** To evaluate OAD tasks, following current SoTA (Pang et al., 2025), we use segment-wise F1 score with an IoU threshold of 0.1 (S-F1) and point-wise F1 score for action start detection with a 1s threshold (P-F1). For step recognition/forecasting and task recognition, we report top-1 accuracy following existing literature (Chen et al., 2024a; Wu et al., 2024).

**Implementation.** We use Llama3 (Dubey et al., 2024) variants as the primary language decoder with a 2-layer MLP adapter, and additionally evaluate Qwen2.5 (Qwen et al., 2025) variants. Task-specific visual encoders are employed following prior works (Wang et al., 2023; Pang et al., 2025; Chen et al., 2024a; Wu et al., 2024). For OAD tasks, we use RGB features at 1 or 4 FPS, with 1 global token per frame. Since OAD requires untrimmed videos, video frames are sampled backwards from the current timestamp within a fixed window to capture both short- and longer-term context (Xu et al., 2021). For step or task recognition, we follow Chen et al. (2024a) and use SigLIP-ViT-L-384 (Zhai et al., 2023) with 2 FPS sampled videos, producing 10 tokens per frame (1 global and 9 patch tokens). Trimmed or full videos are used directly, with downsampling applied only when the sequence exceeds the predefined maximum length. The LLMs are finetuned using LoRA with $r = 128$ and $\alpha = 256$. The balance factor is set to $\lambda = 1$ by default. In line with Chen et al. (2024a), we post-process generative outputs by using Levenshtein edit distance to match the generated text to the closed-set action labels. Additional details are provided in Appendix B.1.

### 4.2 MAIN RESULTS

Experiments are designed to systematically investigate the raised research questions.

**Generative versus Discriminative Classifier.** Tables 1 and 2 compares the performance of generative and discriminative classifiers. The discriminative classifier (Disc) consistently outperforms the generative one (Gen) across datasets and LLM variants. It achieves notable improvements for OAD tasks, especially on EPIC-Kitchens-100, where the large number of fine-grained actions (around 3,600) increases semantic overlap, yielding gains of 6%, and 5% in segment-, and point-wise metrics. The discriminative classifier also surpasses the generative classifier by an average 6%, 3.5%, and 3%

Table 2: Generative (Gen) vs. discriminative (Disc) classifier on COIN for step/task and next step prediction.

| LLM | Model | COIN Benchmark | | |
|---|---|---|---|---|
| | | Step | Next | Task |
| Llama3.2-1B-Instruct | Gen | 57.5 | 45.8 | 90.9 |
| | Disc | **64.1** | **50.1** | **92.8** |
| Llama3-8B-Instruct | Gen | 61.3 | 48.3 | 92.3 |
| | Disc | **66.4** | **51.0** | **94.3** |

Figure 3: False positives for "add sugar". Generative classifier incurs more diverse misclassifications.

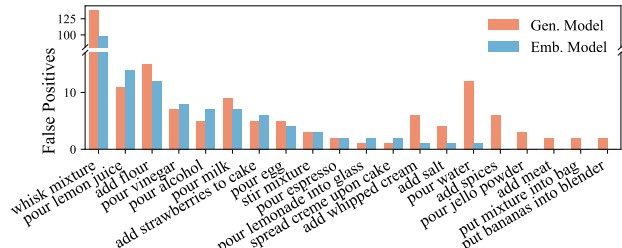

on step recognition, forecasting, and task recognition. Remarkably, the 1B-version discriminative model even outperforms the 8B-version generative model.

We attribute these performance improvements to the discriminative classifier's ability to disregard output semantics. For example, on CrossTask, Fig. 3 shows that, due to the shared verb 'add', the generative classifier produces more false positives (after mapping the generative output to action categories), such as labeling "add sugar" as "add meat" or "add spices". These diverse errors are not observed in the discriminative classifier's predictions.(See Appendix E for more analysis.) Following Sikar et al. (2024), we further introduce an entropy-based diversity score to measure the spread of misclassification, with higher values indicating more diverse errors. The generative classifier scores 0.76, 1.3 and 1.8 on CrossTask, EPIC-Kitchens-100 and Ego4DGoalStep, respectively, compared to 0.66, 0.79 and 1.5 for the discriminative classifier, reflecting the diverse false predictions caused by the introduced semantics, see Appendix B.3.

Discriminative classifiers also offer faster inference compared to generative ones, as it predicts outputs in a single step rather than generating tokens autoregressively. As shown in Tab. 1, the discriminative classifier achieves speedups proportional to the token count in action labels. On Ego4DGoalStep, the discriminative classifier is nearly 4× faster than generative, where the average action label length is 5.5 tokens, compared to 2.1 in THUMOS, 2.6 in CrossTask, and 2.4 in EK100, using the LLaMA3 tokenizer. Although the reported FPS excludes vision encoder computation by using pre-extracted features, the encoder itself runs at a comparable frame rate (Pang et al., 2025), making this speedup a meaningful improvement for real-time applications. Similarly, the discriminative classifier accelerates training by avoiding forward and backward passes through many tokens, achieving approximately 1.8× faster training on OAD tasks.

**Bridging the Performance Gap.** The observed performance gap between the discriminative and generative classifier motivates us to investigate its underlying cause. We observed that this gap should stem from the shared semantics in the generative output space. To verify this, we design three experimental settings in Fig. 4 that progressively disrupt these semantics, aiming to bridge the performance gap between the generative and discriminative classifier.

- Randomized Consistent Mapping. Action labels are split into subwords by the tokenizer. Each subword is replaced with a random token, but identical subwords are mapped to the same token across all labels.
- Desynchronized Independent Mapping: Each subword is replaced independently with a random token, so the same subword will map to different tokens in different labels.
- Extended Vocabulary: Add new action labels to the tokenizer vocabulary and initialize their embeddings by randomly selecting from the original vocabulary. Each action is thus represented by a single newly added token.

The corresponding results of these settings are presented in Tab. 3, highlighting several key observations. First, randomizing tokens removes subword semantics and its relationships with neighbors, yet results remain unchanged, indicating that subword semantics and local word connections are not so critical. This is expected, as fine-tuning on predefined action labels encourages memorization. Additionally, action labels are concise and expressed in a simplified style rather than natural language, making word transitions poorly captured and less important. By further desynchronizing tokens to eliminate overlap among action labels, shared semantics are removed, leading to significant performance gains that approach the discriminative classifier. This result supports our hypothesis that the shared semantics in the generative output space are the key cause of the performance gap.

Table 3: Bridging generative and discriminative classifiers using different tokenization with Llama3.2-1B-Instruct.

| Model | OAD Benchmark (segment-F1 / point-F1) | | |
|---|---|---|---|
| | CrossTask | EPIC-Kitchens-100 | Ego4DGoalStep |
| Gen | 46.8 / 31.7 | 16.7 / 13.9 | 8.9 / 3.4 |
| Gen_rand | 46.9 / 31.8 | 16.8 / 14.0 | 8.8 / 3.5 |
| Gen_desync | 48.7 / **34.1** | 23.0 / 19.0 | 10.4 / 3.9 |
| Gen_extend | **48.9** / 33.9 | **23.3** / 19.2 | 10.5 / 4.0 |
| Disc | 48.8 / 34.0 | 23.2 / **19.3** | **10.6 / 4.1** |

Figure 4: Tokenization strategies. $\alpha$ and $\beta$ denote random tokens.

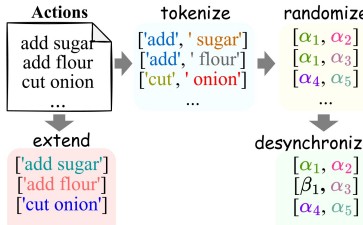

Similarly, extending the tokenizer with action labels ensures non-overlapping tokens and achieves comparable performance. Unlike desynchronized mapping, each action is represented by a single token, allowing prediction in a single pass without token-by-token generation. This could help explain the discriminative classifier as a single-step generation process and align discriminative and generative approaches.

**Generation-Assisted Discriminative Classifier.** While the discriminative classifier are more effective in classification settings, the generative modeling enables free-form generation, producing flexible outputs to support representation learning for classification. We investigate several scenarios to explore how generative capabilities can enhance classifcation performance using the proposed Generation-Assisted Discriminative Classifier.

- Labeling. The generative head produces the target action label as text, performing the same task as the discriminative classifier. It is expected to function as a regularizer, incorporating the label lexical semantics that the discriminative classifier alone does not capture.
- Context generation. The generative classifier generates additional outputs beyond the target labels, such as related past or future actions or the overall video goal. This process is expected to enhance representation learning by providing contextual and temporal cues.

To incorporate label semantics, we test two training strategies for labeling: two-stage and joint training. In two-stage training, the model is first trained using only the generative objective. All parameters are then frozen to preserve the learned semantics, except the [CLS] token and classification head, which are trained in the subsequent stage. As shown in Tab. 4, the two-stage training results in significantly lower performance, suggesting that the generative classifier alone does not provide strong representations. On the other hand, joint training enables representation learning alongside generative responses, but with only slightly performance improvement, further suggesting that lexical semantics have limited impact in a closed-vocabulary setting.

Table 4: Comparison with GAD variants with Llama3.2-1B-Instruct.

| Model | OAD Benchmark (segment-F1 / point-F1) | | | COIN (Top-1 acc.) | | |
|---|---|---|---|---|---|---|
| | CrossTask | EPIC-Kitchens-100 | Ego4DGoalStep | Step | Next | Task |
| **Disc** | 48.8 / 34.0 | 23.2 / 19.3 | 10.6 / 4.1 | 64.1 | 50.1 | 92.8 |
| **GAD** (label_2stage) | 41.8 / 26.7 | 9.3 / 6.7 | 5.9 / 1.7 | 35.9 | 28.0 | 85.5 |
| **GAD** (label_joint) | 49.1 / 33.9 | 23.6 / 19.6 | 10.8 / 4.1 | 64.4 | 50.4 | 93.2 |
| **GAD** (context) | **50.3 / 34.5** | **24.1 / 20.1** | **11.0 / 4.3** | **65.3** | **51.4** | **93.5** |

For context generation, the generation outputs the target label along with auxiliary information. In OAD, where only action labels are available, we explore action relationships as a form of context, specifically by generating the previous action as the auxiliary output to support current predictions. This proves more effective than generating future actions, as the previous action is further supported by visual information from the input video, which contains only past observations in the online setting. More ablations can be found in Appendix B.4. In COIN, where task information is available, we explore the role of task information as auxiliary outputs. Here, no extra information is used, since task label are already included during training, where task recognition and step recognition/forecasting are jointly learned within a single model. Results in Tab. 4 show that both the task and previous action information help, with the task cues having a stronger impact. Using the previous action as auxiliary output brings more benefit to CrossTask, where actions are closely aligned with a single goal, unlike other datasets where actions follow multiple, loosely related tasks. These results suggest that the generative modeling would be more helpful when it generates information complementary to the discriminative classifier.

For efficiency, while incorporating generative objectives slows down training similarly to the generative classifier, inference speed matches that of the discriminative classifier by removing token generation, ensuring efficient inference.

### 4.3 ABLATION STUDIES

We conduct ablation studies on key design choices of our proposed GAD, evaluated on OAD tasks with results reported in Tab.5. More ablations are included in Appendix B.4.

- The learnable token. We adopt a learnable [CLS] token for discriminative prediction. An alternative is the last visual token, which also aggregates all preceding visual and text information. Results (w/o [CLS]) show that using [CLS] token performs better, likely due to its stronger generalization ability, while the last visual token tends to overfit the training data.
- Auxiliary task. To confirm the benefits of capturing semantics via generation, we convert previous-action prediction into a standalone classification objective, adding an extra classifier that uses either the existing learnable token (GAD_prev_disc) or an additional learnable token (GAD_prev_disc+). Results indicate that neither variant improves learning, and both can harm current-action accuracy, indicating that the advantage arises from generative semantic encoding rather than the auxiliary classification task itself.

Table 5: Ablation studies of our GAD classifier on OAD tasks.

| Model | OAD Benchmark. (frame acc./segment F1/point F1) | | |
| --- | --- | --- | --- |
| | CrossTask | EPIC-Kitchens-100 | Ego4DGoalStep |
| **Disc** | 81.7/48.8/34.0 | 34.8/23.2/19.3 | 33.9/10.6/4.1 |
| **GAD** | 81.8/50.3/34.5 | 35.1/24.1/20.1 | 34.4/11.0/4.3 |
| w/o [CLS] | 81.5/49.2/33.8 | 34.6/20.7/17.7 | 33.7/8.6/3.6 |
| **GAD**_prev_disc | 80.8/48.2/31.6 | 34.8/23.6/19.3 | 33.2/10.1/3.8 |
| **GAD**_prev_disc+ | 80.9/49.5/32.7 | 34.7/23.0/19.6 | 33.3/10.6/4.0 |

### 4.4 COMPARISON WITH SOTA.

We benchmark our approach against state-of-the-art methods in Tables 6 and 7. Although not tailored to any specific task, our approach still achieves state-of-the-art performance. Further gains can be expected through task-specific adaptations. To ensure a fair comparison, we use the same visual features as state-of-the-art methods, showing that our gains arise from the architectural design rather than from incorporating advanced features. For online action detection, we present the first LLM-based method, significantly improving segment- and point-wise performance. For step/task recognition and step forecasting, our approach also largely outperforms recent methods. Notably, our 1B model even surpasses prior 8B models, highlighting the effectiveness of the discriminative classifier augmented with generative assistance.

### 4.5 QUALITATIVE RESULTS.

Fig. 5 presents qualitative results on the COIN benchmark. The generative classifier (Gen) likely generates semantically similar but incorrect outputs, while the discriminative classifier (Disc) demonstrate better discrimination. Our GAD model enhances Disc by generating contextual information that enables task-aware predictions. In the middle example, Disc outputs a task-irrelevant action, while GAD accurately identifies the correct action. We also observe some edge cases where the

Table 6: Comparison with SOTA on OAD tasks.

| Model | **OAD Benchmark** (segment-F1 / point-F1) | | | |
| --- | --- | --- | --- | --- |
| | THUMOS'14 | CrossTask | EPIC-Kitchens-100 | Ego4DGoalStep |
| **Testra** (Zhao & Krähenbühl, 2022) | 43.0 / 32.0 | 48.4 / 33.8 | 16.5 / 14.7 | 8.7 / 3.5 |
| **MAT** (Wang et al., 2023) | 49.4 / 34.2 | 49.7 / 34.2 | 17.5 / 15.5 | 9.5 / 3.8 |
| **CMeRT** (Pang et al., 2025) | 48.9 / 34.6 | 50.1 / 34.3 | 17.7 / 15.8 | 9.7 / 3.9 |
| **Disc** (Llama3.2-1B-Instruct) | 57.8 / 40.1 | 48.8 / 34.0 | 23.2 / 19.3 | 10.6 / 4.1 |
| **GAD** (Llama3.2-1B-Instruct) | **58.1 / 40.2** | **50.3 / 34.5** | **24.1 / 20.1** | **11.0 / 4.3** |

Table 7: Comparison with SOTA on COIN Benchmark.

| Model | COIN Benchmark (Top-1 acc.) | | |
|---|---|---|---|
| | Step | Next | Task |
| **VideoTaskGraph** (Ashutosh et al., 2023) | 57.2 | 40.2 | 90.5 |
| **Videollm-online-8B** (Chen et al., 2024a) | 63.1 | 49.1 | 92.7 |
| **Videollm-MoD-8B** (Wu et al., 2024) | 63.4 | 49.7 | 92.8 |
| **StreamMind-8B** (Ding et al., 2025) | 63.7 | 49.9 | 93.2 |
| **Disc** (Llama3.2-1B-Instruct) | 64.1 | 50.1 | 92.8 |
| **GAD** (Llama3.2-1B-Instruct) | **65.3** | **51.4** | **93.5** |
| **Disc** (Llama3-8B-Instruct) | 66.4 | 51.0 | 94.3 |
| **GAD** (Llama3-8B-Instruct) | **67.3** | **51.6** | **94.5** |

model predicts correctly despite flawed generations. Commonly, the generation identifies the task but answers the query incorrectly, or is completely wrong or misformatted. These edges cases highlight the benefit of performing generation after the learnable token. The learned representation remains robust to generative errors while still benefiting from generative regularization. More analysis is provided in Appendix B.6 and B.5.

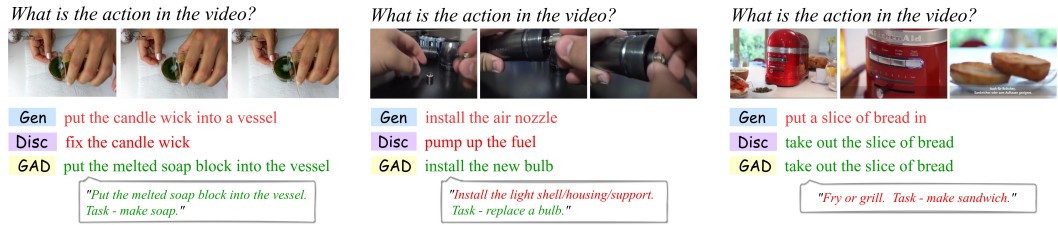

Figure 5: Qualitative results on the step recognition for Generative (Gen) vs. discriminative (Disc) classifier. Red text denotes false predictions and green correct predictions. Callouts display GAD generation outputs.

### 4.6 GENERALIZATION TRADE-OFF

MLLMs are naturally strong in open-world scenarios due to their broad, data-driven pretraining, but perform poorly in closed-set action understanding (see zero-shot performance in Appendix C). Our classification setting involves task-specific fine-tuning, which boosts action-classification performance but reduces general MLLM capabilities, reflecting task-induced forgetting well-documented in the literature (Oh et al., 2024; Han et al., 2024). We provide extra experiments to evaluate generalization and show this trade-off in Appendix D. In fact, the generative head in our GAD model enables leveraging self-curated instruction-tuning data to preserve generalization, *e.g.* augmenting action labels. In addition, since the backbone architecture remains unchanged, removing the introduced LoRA adapters can fully restore the model's open-world performance. We leave a more comprehensive exploration of the trade-off between fine-tuning and generalization to future work.

## 5 CONCLUSION

This paper studies fine-tuning multimodal large language models for downstream temporal action understanding. We identify the fundamental limitations of generative modeling in classification tasks, showing how semantic label overlap constrains performance, and demonstrate the advantages of a discriminative formulation. Building on these insights, we introduce a unified generative-assisted discriminative (GAD) classifier that leverages generative modeling as an auxiliary objective to enhance discriminative learning. Crucially, our approach preserves full compatibility with pretrained models, requiring no changes to the pretraining process.

**Limitations.** While our generative-assisted discriminative classifier performs strongly in closed-set scenarios, discriminative models are still limited to the closed set and cannot directly handle novel or unseen actions. Additionally, task-specific fine-tuning can hurt general-purpose abilities like question answering. Future work could focus on improving generalization to new classes through generative components, while preserving the strengths of discriminative learning for the closed set.

ACKNOWLEDGMENT

This research is supported by the Ministry of Education, Singapore, under the Academic Research Fund Tier 1 (FY2025).

ETHICS STATEMENT

The methods, data, and results presented in this paper raise no known ethical concerns. All experiments were conducted using publicly available datasets under their respective licenses. This paper also did not involve ethically sensitive activities, such as research with human subjects, dataset releases, potentially harmful experiments, or issues related to discrimination & privacy.

REPRODUCIBILITY STATEMENT

All technical details, such as experiment settings, evaluation protocols and implement instructions are detailed described in Sec.4.1 of the paper and Sec.B.1 of the appendix to ensure reproducibility. All datasets and code used in this paper are publicly accessible.

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

APPENDIX

THE USE OF LARGE LANGUAGE MODELS

In this paper, Large Language Models (LLMs) were used exclusively for polishing the manuscript, including improving writing style, enhancing readability, and correcting grammatical errors. LLMs were not employed for research purposes such as literature retrieval, idea generation, or discovery. All methodological proposals, experimental designs, analyses, and conclusions were developed without the involvement of LLMs.

## A   FEATURE ANALYSIS OF GENERATIVE VS. DISCRIMINATIVE CLASSIFIER

We examine the feature quality of the generative and discriminative classifiers. For generative outputs, action labels are tokenized into multiple tokens, and representations are derived using one of four strategies: mean, max, first, or last token. Full t-SNE visualizations reveal the clear decision boundaries of the discriminative classifier, further highlighting the comparatively inferior performance of generative decoding.

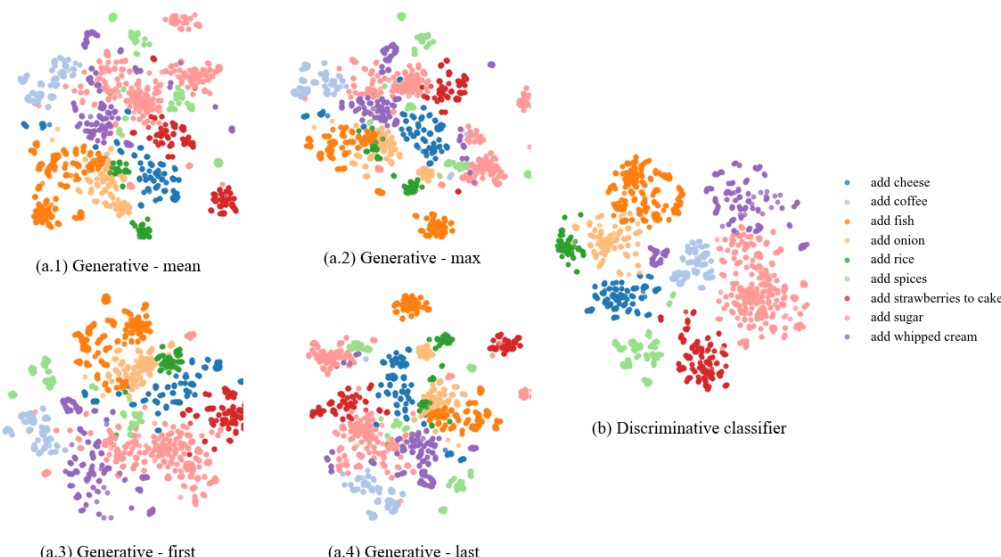

Figure 6: T-SNE plot comparing feature spaces of the generative and discriminative classifier on CrossTask dataset for actions sharing the verb 'add'. Generative feature is derived from token representations using four strategies: mean, max, first, or last token. Discriminative feature is the representation of the learnable token.

## B   EXPERIMENTS

### B.1   IMPLEMENTATION DETAILS

We conduct evaluations on two benchmarks: OAD, for online action detection, and COIN, for step/task recognition and step forecasting. The used task-specific queries are summarized in Tab. 8.

For the OAD benchmark, training involves sampling a "current" timestamp $t$, and long- and short-term memories are constructed by padding or cropping past observations up to $t$. The timestamp is sampled using a sliding window with a random start and a stride equal to the short-term length, while frames in short- and long-term are selected at a fixed predefined sample rate. During inference, an online streaming setup is simulated with a sliding window of stride 1 and fixed start time at 0, predicting one frame at a time. We train for 50 epochs on THUMOS'14, CrossTask, and EPIC-Kitchens-100,

Table 8: Task-specific queries.

| Task | Query |
|---|---|
| Online action detection | *What is the action in the last frame?* |
| Step recognition | *What is the action in the video?* |
| Step forecasting | *What is the next action in the video?* |
| Task recognition | *What is the overall activity in the video?* |

and 20 epochs on Ego4DGoalStep, where the latter contains fewer distracting background frames, making it easier to train. As training samples are generated with a stride equal to the short-term length, only a fraction of frames is used per epoch. For example, 50 epochs on CrossTask with stride 20 is equivalent to 2.5 full passes over the dataset. Optimization is performed with AdamW using a learning rate of 0.0001, a warmup ratio of 0.1, and a batch size of 32 across all datasets. Unless otherwise specified, we employ Llama3.2-1B-Instruct as the language model. Details of the hyper-parameters for OAD can be found in Tab. 9.

Table 9: Hyperparameters for OAD setting.

| Datasets | long-term (s) | short-term (s) | sample rate | visual feature@FPS |
|---|---|---|---|---|
| THUMOS'14 | 128 | 32 | 1 | ResNet50@4 |
| CrossTask | 128 | 20 | 1 | DINOV2@1 |
| EPIC-Kitchens-100 | 128 | 20 | 1 | TSN@4 |
| Ego4DGoalStep | 128 | 16 | 1 | DINOV2@1 |

For the COIN benchmark, we adopt a multi-task learning framework to jointly train step recognition, next-step forecasting, and task recognition using task-specific queries, with separate MLP heads for step and task predictions. Training is conducted for 5 epochs using the AdamW optimizer with a learning rate of 0.0001, a warmup ratio of 0.05, and a batch size of 8.

**Metric**: For OAD tasks, we report only segment-wise and point-wise F1 scores, omitting frame-wise performance, since frame-level accuracy may not provide sufficiently informative or reliable evaluation, as shown in Pang et al. (2025). For completeness, frame-wise accuracy results are included in the supplementary material.

## B.2    GAD UNIFICATION STRATEGY

The generative and discriminative classifiers can be unified either sequentially or in parallel. We investigate three unification strategies, illustrated with representative examples.

- Sequential learning with discriminative learning first. The learnable `[CLS]` token is first processed, serving as a conditioning context for the generative output.

  <|begin_of_text|>
  User: What is the action in the last frame?
  \<v\>\<v\>\<v\>\<v\>\<v\>\<v\>\<v\>\<v\>\<v\>\<v\>\<v\>\<v\>\<v\>\<v\>\<v\>\<v\>\<v\>\<v\>\<v\>\<v\>\<v\>\<v\>\<v\>`[cls]`
  Assistant: flip bread \<eos\>

- Sequential learning with generation first. The learnable `[CLS]` token is positioned after generation, enabling the discriminative learning to be conditioned on the generative outputs.

  <|begin_of_text|>
  User: What is the action in the last frame?
  \<v\>\<v\>\<v\>\<v\>\<v\>\<v\>\<v\>\<v\>\<v\>\<v\>\<v\>\<v\>\<v\>\<v\>\<v\>\<v\>\<v\>\<v\>\<v\>\<v\>\<v\>\<v\>\<v\>
  Assistant: flip bread \<eos\>`[cls]`

- Parallel learning. Discriminative learning and generative decoding are performed concurrently in two branches, sharing the same text and visual tokens, but without conditioning on one another.

To fairly assess the impact of unification strategies, we align the generative task with discriminative learning in GAD by generating only the target label, as generating extra previous action could

```
<|begin_of_text|>
User: What is the action in the last frame?
<v><v><v><v><v><v><v><v><v><v><v><v><v><v><v><v><v><v><v><v><v><v><v><v><v><v><v>[cls]

<|begin_of_text|>
User: What is the action in the last frame?
<v><v><v><v><v><v><v><v><v><v><v><v><v><v><v><v><v><v><v><v><v><v><v><v><v><v><v>
Assistant: flip bread <eos>
```

Table 10: Discriminative and generative performance in the proposed GAD classifier, where generative modeling is for labeling task same as the discriminative learning.

| Model | OAD Benchmark. (frame acc./segment F1/point F1) | | |
| | CrossTask | EPIC-Kitchens-100 | Ego4DGoalStep |
|---|---|---|---|
| **Gen** | 81.3/46.8/31.7 | 32.9/16.7/13.9 | 32.9/8.9/3.4 |
| **Disc** | 81.7/48.8/34.0 | 34.8/23.2/19.3 | 33.9/10.6/4.1 |
| **GAD**_gen | 81.5/46.8/31.8 | 32.1/16.0/13.2 | 31.4/7.8/3.0 |
| **GAD**_emb | 81.6/49.1/33.9 | 34.8/23.6/19.6 | 34.0/10.8/4.1 |
| **GAD**_seq-g_gen | 81.2/46.3/31.3 | 33.3/17.1/14.2 | 32.5/8.9/3.3 |
| **GAD**_seq-g_emb | 81.2/46.3/31.3 | 33.3/17.1/14.2 | 31.5/8.7/3.4 |
| **GAD**_parallel_gen | 81.5/46.9/31.8 | 32.4/16.1/13.4 | 32.2/8.4/3.1 |
| **GAD**_parallel_emb | 81.7/49.4/34.1 | 34.9/23.6/19.6 | 34.1/10.7/4.1 |

interfere with current action prediction and bias performance. From the results in Tab. 10, we observe the conflicts between generation and discriminative learning in sequential learning setting. In the discriminative learning first strategy (GAD), discriminative predictions are largely unaffected and can even benefit slightly from semantic regularization through generation, whereas generation performance suffers when conditioned on the learned token. In contrast, in the generation first strategy (GAD_seq-g), generation performance remains less affected, but discriminative performance degrades, sometimes exactly matching the generative one. This happens because conditioning on generative outputs enables the discriminative classifier to learn a shortcut to aligns its representation with the generative outputs during training, causing it to effectively replicate the generative output at inference. Since generative outputs are more prone to confusion due to semantic overlap, which can in turn degrade discriminative classifier performance. Meanwhile, placing the learnable token at the end slows inference, as autoregressive generation must be completed first.

In the parallel learning strategy (GAD_parallel), discriminative and generative performance are less affected by interference. Training them simultaneously is analogous to multi-task learning with a shared backbone, allowing the model to capture cross-task knowledge while benefiting from regularization. However, this strategy needs two forward passes for each output, which reduces training efficiency.

In conclusion, given our focus on discriminative outputs, we stick with the discriminative learning first strategy, which yields strong representations, exploits generative regularization, and ensures efficient training and inference.

### B.3 ENTROPY-BASED DIVERSITY SCORE FOR MISCLASSIFICATIONS

Since the generative classifier treats action labels as subword tokens, shared subwords across similar actions can lead to greater confusion. To quantify this effect, we introduce an entropy-based Diversity Score (DScore) to measure the variability of misclassified predictions. We first compute the confusion matrix $C \in \mathbb{R}^{N \times N}$ of a given classifier, where $N$ denotes the number of classes, with rows representing ground truth labels and columns representing predictions. Then, we set the diagonal entries (true positives) of the confusion matrix to zero to focus on misclassifications. For OAD datasts, we additionally exclude misclassifications assigned to the background class to avoid diluting the results, as background is semantically unrelated to other classes and overwhelmingly represented in the datasets. Finally, we normalize each row of the modified confusion matrix $C'$ to obtain an error distribution $p$ for each target class, and quantify the diversity of misclassifications for each class

using the Shannon entropy $H$. The final DScore is computed as the average entropy over actions, excluding the background and classes absent at test time.

$$p_{i,j} = \frac{C'_{i,j}}{\sum_{k=0}^{N-1} C'_{i,k}}, \quad H_i = -\sum_{k=0}^{N-1} p_{i,j} \log(p_{i,j} + \epsilon) \qquad \forall i, j \in [0, N-1], \tag{4}$$

where $\epsilon$ is a small constant ensuring valid input to the logarithm.

Table 11 presents diversity scores on the OAD datasets, showing that the generative classifier produces more diverse misclassifications due to semantic overlap. In contrast, our GAD model can encode semantics through generation while mitigating the error diversity, leading to more consistent predictions and thus potentially reducing action over-segmentation.

Table 11: Comparison of misclassification diversity for generative (Gen), discriminative (Disc), and generation-assisted discriminative (GAD) classifiers.

| Model | OAD Benchmak. DScore | | |
|---|---|---|---|
| | CrossTask | EPIC-Kitchens-100 | Ego4DGoalStep |
| Gen | 0.76 | 1.3 | 1.8 |
| Disc | 0.66 | 0.79 | 1.5 |
| GAD | 0.67 | 0.80 | 1.49 |

### B.4 ABLATION STUDIES

We conduct ablation studies on key design choices of our proposed GAD, evaluated on OAD tasks with results reported in Tab.12.

- The generative output can be any text, but we focus on generating the previous action. Beyond the flexibility of free-form outputs, we aim to show that the benefits stem from generation itself rather than the auxiliary task. To this end, we reformulate previous action prediction as a separate classification task, introducing an additional classifier that uses either the representation of the existing learnable token (GAD_prev_disc) or a new learnable token appended after it (GAD_prev_disc+). Results indicate that neither approach improves learning and can even disrupt current action classification, causing a performance drop. This further highlights the advantage of using generation to encode semantics as an auxiliary task.
- We use the generative head for previous step generation. Alternatives include generating the next action or past actions within a fixed short-term window. Results show that these alternatives still outperform the discriminative baseline by capturing action relationships, but remains less effective than previous step generation. Next-step generation lacks corresponding visual inputs, while generating multiple past actions reduces performance due to increased complexity that diverts focus from discriminative learning.
- For discriminative outputs, we use the representation of the learnable `[CLS]` token for prediction. An alternative is to use the last visual token, which also aggregates all preceding visual information. This design fits the setting of online action detection, which requires recognizing the action in the last frame. However, results (w/o `[CLS]`) show that using `[CLS]` token performs better, likely due to its stronger generalization ability, while the last visual token tends to overfit the training data.
- The generative response is designed to include the target label along with the context information describing the previous action. Therefore, generative and discriminative learning are somehow aligned toward predicting the same target action. To evaluate the effect of separating the tasks, we remove the target label from the generative output, leaving only the context generation (GAD_sep). While GAD_sep still performs well, it underperforms the original version, suggesting that sharing the same target better aligns the discriminative and generative modules, while also providing valuable context for the action relationship learning.

### B.5 ANALYSIS ON GAD GENERATION OUTPUT

We use generation as an auxiliary task and rely solely on the discriminative output during inference. Nevertheless, it is important to examine the quality of the generative outputs and their role in

Table 12: Ablation studies of our GAD classifier on OAD tasks.

| Model | OAD Benchmark. (frame acc./segment F1/point F1) | | |
|---|---|---|---|
| | CrossTask | EPIC-Kitchens-100 | Ego4DGoalStep |
| **Disc** | 81.7/48.8/34.0 | 34.8/23.2/19.3 | 33.9/10.6/4.1 |
| **GAD** | 81.8/50.3/34.5 | 35.1/24.1/20.1 | 34.4/11.0/4.3 |
| **GAD**_prev_disc | 80.8/48.2/31.6 | 34.8/23.6/19.3 | 33.2/10.1/3.8 |
| **GAD**_prev_disc+ | 80.9/49.5/32.7 | 34.7/23.0/19.6 | 33.3/10.6/4.0 |
| **GAD**_next | 81.6/49.8/34.3 | 34.6/23.7/19.8 | 34.1/10.8/4.2 |
| **GAD**_past | 81.7/49.5/34.1 | 34.7/23.9/19.9 | 33.9/10.4/4.1 |
| w/o `[CLS]` | 81.5/49.2/33.8 | 34.6/20.7/17.7 | 33.7/8.6/3.6 |
| **GAD**_sep | 81.5/50.0/33.9 | 34.7/24.0/20.1 | 34.3/10.9/4.3 |

enhancing the discriminative learning. To this end, we specifically analyze the generation results. We conduct this analysis on the COIN benchmark, as its accuracy metric provides a clearer basis for evaluation.

We analyze the correctness of three types of GAD output: the discriminative output (Disc), the generation output for the target label (Gen), and the generation output for the task label (Gen_extra). These yield eight possible output combinations for step recognition and forecasting, as illustrated in Fig. 7. In the case of task recognition, the target label and task label are identical, so Gen_extra is not applicable, leading to four possible output combinations.

By examining the distribution of output correctness combinations, we make several observations. First, the three types of outputs are generally consistent, with most cases fully correct. Second, step generation is more challenging than task, as tasks exhibit clearer separation with lower semantic overlap. As such, cases where the task generation is incorrect while the step generation is correct are uncommon. Third, there are instances where both generative outputs are flawed, yet the discriminative output remains accurate, highlighting the benefit of placing the discriminative learning ahead of generation to avoid the impact of erroneous generated content. Although the discriminative learning is not conditioned on generation, it is still influenced by generative outputs, which regularize it to be generation-aware. Finally, some cases show the generative outputs are accurate while the discriminative output does not, highlighting complementarity between the two and suggesting potential gains through ensembling.

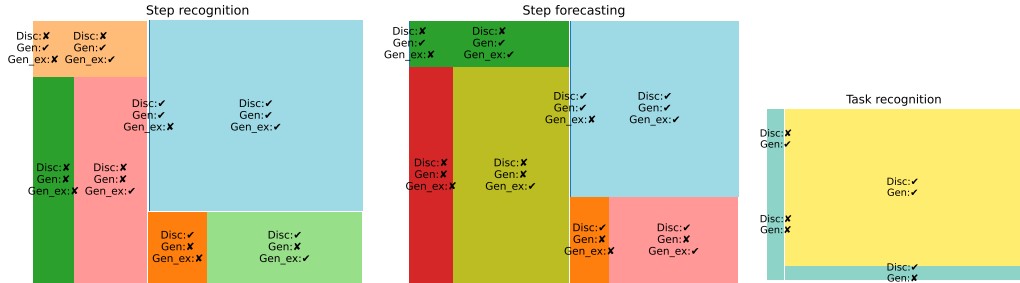

Figure 7: Quantitative analysis of discriminative ouputs (Disc) and generative outputs (Gen & Gen_ex). Larger areas indicate higher occurrence. Checks and crosses denote correct and incorrect outputs, respectively.

## B.6 QUALITATIVE RESULTS

We provide additional qualitative results on the OAD task and COIN benchmark in Fig. 8. We observe that even when the previous step generation is occasionally incorrect, the relationship between the generated previous step and the current step remains meaningful, *e.g.* 'close cupboard' following 'open cupboard'. This demonstrates that the model can captures action relationships through generation. Similarly to the COIN benchmark, we also observe that the generative and discriminative

Table 13: Comparison with SOTA on OAD tasks.

| Model | OAD Benchmark. (frame acc./segment F1/point F1) | | | |
| --- | --- | --- | --- | --- |
| | THUMOS'14 | CrossTask | EPIC-Kitchens-100 | Ego4DGoalStep |
| **Testra** (Zhao & Krähenbühl, 2022) | 76.9/43.0/32.0 | 81.2/48.4/33.8 | 34.7/16.5/14.7 | 31.7/8.7/3.5 |
| **MAT** (Wang et al., 2023) | 78.4/49.4/34.2 | 81.4/49.7/34.2 | 35.0/17.5/15.5 | 32.9/9.5/3.8 |
| **CMeRT** (Pang et al., 2025) | 78.2/48.9/34.6 | 81.3/50.1/34.3 | **35.2**/17.7/15.8 | 32.8/9.7/3.9 |
| **Gen** (Llama3.2-1B-Instruct) | 78.6/56.9/38.8 | 81.3/46.8/31.7 | 32.9/16.7/13.9 | 32.8/8.9/3.4 |
| **Disc** (Llama3.2-1B-Instruct) | 78.5/57.8/40.1 | 81.7/48.8/34.0 | 34.8/23.2/19.3 | 33.9/10.6/4.1 |
| **GAD** (Llama3.2-1B-Instruct) | **78.8/58.1/40.2** | **81.8/50.3/34.5** | 35.1/**24.1/20.1** | **34.4/11.0/4.3** |

Table 14: Comparison with SOTA on OAD tasks.

| Model | **OAD Benchmark** (segment-F1 / point-F1) | | |
| --- | --- | --- | --- |
| | CrossTask | EPIC-Kitchens-100 | Ego4DGoalStep |
| **Gen** | 46.8 / 31.7 | 16.7 / 13.9 | 8.9 / 3.4 |
| **Testra** [1] | 48.4 / 33.8 | 16.5 / 14.7 | 8.7 / 3.5 |
| **MAT** [2] | 49.7 / 34.2 | 17.5 / 15.5 | 9.5 / 3.8 |
| **CMeRT** [3] | 50.1 / 34.3 | 17.7 / 15.8 | 9.7 / 3.9 |
| **Disc** | 48.8 / 34.0 | 23.2 / 19.3 | 10.6 / 4.1 |
| **Gen_rand** | 46.9 / 31.8 | 16.8 / 14.0 | 8.8 / 3.5 |
| **Gen_desync** | 48.7 / 34.1 | 23.0 / 19.0 | 10.4 / 3.9 |
| **Gen_extend** | 48.9 / 33.9 | 23.3 / 19.2 | 10.5 / 4.0 |
| **GAD** | **50.3 / 34.5** | **24.1 / 20.1** | **11.0 / 4.3** |

outputs are not always consistent, highlighting their complementary nature and motivating further analysis of their ensemble performance.

### B.7 SOTA COMPARISON ON OAD TASKS

We include additional frame-wise accuracy results for a more comprehensive assessment. The results show significant improvements of our GAD classifer in segment- and point-wise performance while maintaining competitive frame-level accuracy.

## C ZERO-SHOT ACTION UNDERSTANDING

We evaluate two MLLMs zero-shot, Qwen2.5-VL-7B (Bai et al., 2025) and VideoLLM-online (Chen et al., 2024a), on the COIN dataset ( 750 actions). For Videollm-Online model, a procedural video understanding MLLM, we use it for open-ended generation. For Qwen2.5-VL, action prediction is treated either as an open-ended generation task, or by providing action labels as candidate options in the prompt. Post-processing is always applied to match the generation to action categories using the CLIP text encoder and cosine similarity.

As shown in Table 15, Qwen2.5 performs best but still significantly trails our fine-tuned method. These results suggest that action understanding with large, semantically similar actions is challenging, making fine-tuning essential. In addition, providing action categories in the prompt worsens the performance, suggesting that including a large candidate set in the prompt can hinder MLLMs' ability to correctly interpret instructions.

## D GENERALIZATION-PERFORMANCE TRADE-OFF

We prove the necessity of fine-tuning in Sec. C. However, extending fine-tuning boosts performance, but increases memorization and reduces generalization. To quantify memorization, we define the

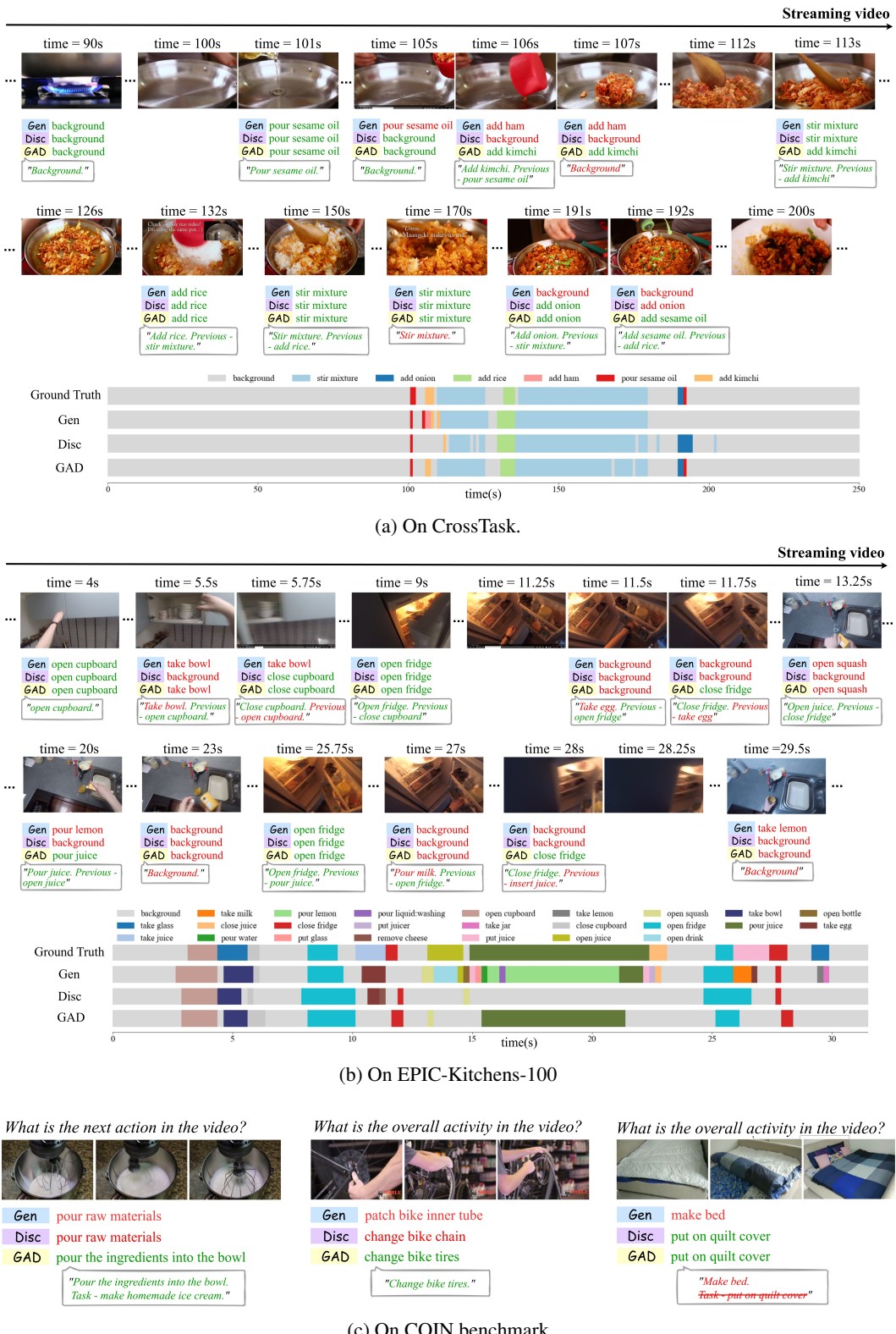

Figure 8: Qualitative results on the OAD and step recognition tasks. Gen - generative classifier, Disc- discriminative classifier. Text in red denotes false predictions, while in green represents correct predictions. Callouts display the generation outputs produced by our GAD. Bar charts in the bottom show predictions for streaming videos.

Table 15: Zero-shot performance of existing MLLMs on COIN dataset.

| Model | Step | Next |
|---|---|---|
| GAD (fine-tuned) | 67.3 | 51.6 |
| Videollm-online-8B (open-ended) | 4.8 | 3.5 |
| Qwen2.5-VL-7B (open-ended) | 16.1 | 8.9 |
| Qwen2.5-VL-7B (categories in prompt) | 11.9 | 6.5 |

memorization rate as the proportion of generated outputs that match training action categories during inference. As shown in Table 16, achieving optimal performance on COIN task recognition (Tang et al., 2019) requires 4 epochs of fine-tuning, yet the memorization rate evaluated on the unseen Breakfast dataset (Kuehne et al., 2014) exceeds 98% after just one epoch, and the zero-shot performance on unseen dataset Breakfast decreases over the course of training. This shows the inherent trade-off between optimal performance and generalization ability.

In fact, the generative head in our GAD model enables leveraging self-curated instruction-tuning data to preserve generalization—for example, by augmenting action labels. However, this comes at the cost of some closed-set performance. We leave a more thorough exploration of these generalization capabilities to future work.

Table 16: Performance-generalization trade-off on our generative baseline.

| | Epoch 0 | Epoch 1 | Epoch 2 | Epoch 3 | Epoch 4 | Epoch 5 |
|---|---|---|---|---|---|---|
| Memorization_ratio (on Breakfast) | 0.0 | 98.1 | 99.6 | 99.8 | 99.9 | 99.9 |
| Accuracy on COIN (test set) | 8.3 | 77.0 | 85.3 | 91.3 | 92.8 | 92.7 |
| Accuracy on Breakfast (unseen) | 6.3 | 4.5 | 3.5 | 3.6 | 3.6 | 3.6 |

# E ERROR ANALYSIS: GENERATIVE VS. DISCRIMINATIVE

We provide extra analysis why the generative baseline produce more diverse misclassifications than the discriminative one. In Figure 9, we compare the confusion matrix comparison on CrossTask for actions shared verb 'add'. The generative baseline exhibits greater difficulty handling semantic overlap among these actions.

Figure 9: Confusion matrix comparison on CrossTask for actions shared verb 'add'. Generative classifier incurs more diverse misclassifications.

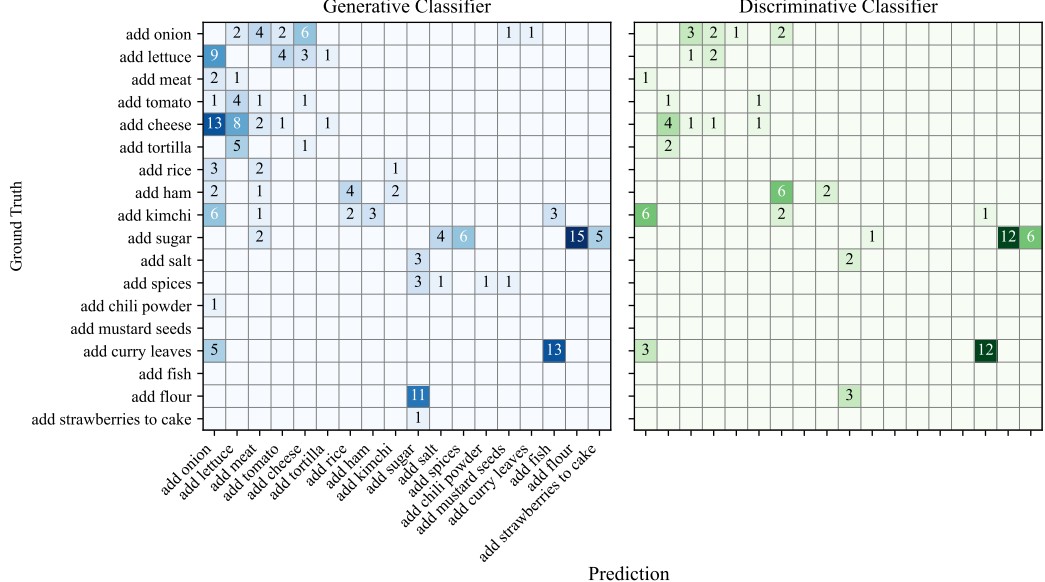

