# OpenReview forum: "On Discriminative vs. Generative classifiers: Rethinking MLLMs for Action Understanding"
_ICLR.cc/2026/Conference — ICLR 2026 Poster_

### Official Review · Reviewer_4Sdb · 2025-10-30

**Soundness:** 2
**Presentation:** 3
**Contribution:** 2
**Rating:** 6
**Confidence:** 4

**Summary:**

Authors explore how Multimodal LLMs (MLLMs) can be used for closed-set action classification, comparing generative and discriminative approaches. They claim that generative classifiers (autoregressively producing text) are less efficient and accurate compared to discriminative classifiers.
The authors proposed a Generation-Assisted Discriminative (GAD) classifier that combined strengths of both aspects. This model uses a discriminative framework for efficient inference but incorporates generative modeling as an auxiliary task during fine-tuning to enhance the model's semantic understanding. Evaluation across five datasets establish clear performance gains in terms of both accuracy and inference speed.

**Strengths:**

1. Strong results across multiple datasets with multiple baselines with focus on both accuracy and speed.
2. Interesting dual training strategy.
3. Clear presentation of key idea (e.g. great use of diagrams in Figure 2).

**Weaknesses:**

1. "approximately 1.8× faster training": Discuss this more - how is the training faster than next-token prediction? This is unclear.

2. Lost generality: does this classification training remove the general QnA abilities of these VLMs? This is unclear.

3. How do these VLMs perform zero-shot (generative classifier) on these tasks? This will provide an interesting point of comparison.

4. Inference compute scaling: if you generate one or two tokens (with the generative part of GAD), and then run the classifier on all tokens, does this improve performance? This will be an interesting analysis.

5. How does cls head scale with more output categories? For example, say 20,000 categories? What is the impact on inference speed?

6. Considering discussing prior work exploring similar ideas for video QnA and NLP. Two examples below.
  - https://arxiv.org/abs/2403.16998 (using MLLMs for closed-set video classification)
  - https://arxiv.org/abs/2210.12353 (using LLMs for closed-set classification)

**Questions:**

See weaknesses.

---

> ### Author Response · Authors · 2025-11-21
> **Author Response (part 1 of 3)**
>
> Dear reviewer 4Sdb
>
> We greatly thank you for the insightful comments, and are glad that our thorough evaluation, dual training strategy, and clear presentation are recognized. We address your raised questions below.
>
> ### **` Q1. 1.8× faster training than next-token prediction`**
>
> Thanks for pointing out the confusion, we will fix it in the update.
> Training with a discriminative classifier is faster than autoregressive token prediction because
> - Token by token prediction increases the input sequence length and requires gradients to backpropagate through every token, thus greatly increasing computation.
> - The classification head is much lighter than the language-modeling head (100~3000 action categories vs. 30k token  vocabulary). But this is a minor factor compared to the autoregressive prediction.
>
>
> ### **`Q2. Impact on general QnA abilities`**
>
> Thank you for raising this important point. Classification training is a task-specific fine-tuning of the VLM. It enables strong action-classification performance but simultaneously reduces general VLM capabilities, including QA. This behavior is a form of task-induced forgetting and is consistent with effects documented in the literature on other tasks [1,2].
> - We note that zero-shot MLLMs struggle with action understanding (see zero-shot results in Q3), which emphasises the need for fine-tuning.
> - We show the gradual trade-off in Table A.  Over the course of fine-tuning, the performance on the corresponding test set (COIN task recognition) improves, while predictions on the unseen Breakfast dataset increasingly collapse into training COIN categories, and its zero-shot QA performance decreases. Balancing task-specific fine-tuning with general QA preservation is interesting, but as we mainly focus on action understanding, we leave this to future work.
> Our method adopts parameter-efficient fine-tuning, where the model could stay general by removing the newly trained task-specific adapter.
>
> [1] Towards Calibrated Robust Fine-Tuning of Vision-Language Models, NeurIPS, 2024
>
> [2] Anchor-based Robust Finetuning of Vision-Language Models, CVPR, 2024
>
>
> *Table A. Performance-generalization trade-off on our generative baseline. The model is trained on COIN task recognition and evaluated on both COIN and unseen Breakfast. Memorization rate is defined as the proportion of generated outputs matching training action categories during inference.*
> | Epoch (iteration)           | Memorization Ratio (Breakfast)               |  COIN Accuracy (test set)      | Breakfast Accuracy(unseen) |
> |------------------------|-----------------------------------|-------------------------------------|-----------------------------------|
> | 0 (0)                    | 0.0                  | 8.28               | 6.33                              |
> | 1 (521)                | 98.1                 | 77.0               | 4.45                              |
> | 2 (1042)               | 99.6                 | 85.3               | 3.56                              |
> | 3 (1563)               | 99.8                 | 91.3               | 3.59                              |
> | 4 (2084)               | 99.9                 | 92.8               | 3.63                              |
> | 5 (2600)               | 99.9                 | 92.7               | 3.62                              |

---

> > ### Comment · Reviewer_4Sdb · 2025-11-23
> > **Unclear**
> >
> > Authors Note:
> > ```
> > Training with a discriminative classifier is faster than autoregressive token prediction because:
> >   - Token by token prediction increases the input sequence length and requires gradients to backpropagate through every token, thus greatly increasing computation.
> > ```
> >
> > 1. "Token by token prediction increases the input sequence length"
> >   - How so? If you refer to teacher forcing, that is the same as batching, and your will be less efficient?
> >   - Could you clarify the exact inputs vs outputs for standard training vs yours?
> >
> > 2. "requires gradients to backpropagate through every token"
> >   - If I understand correctly, gradients need to backpropogate through every token for both cases
> >   - Which exact tokens require gradients in regular training that your training does not need?
> >
> > 3. "The classification head is much lighter than the language-modeling head"
> >   - This is clear, and makes sense
> >   - The 1.8 speedup could be coming from here entirely. If that is the case, that makes more sense.

---

> > > ### Comment · Reviewer_4Sdb · 2025-11-24
> > > **Checking on this again**
> > >
> > > @Authors?

---

> ### Author Response · Authors · 2025-11-21
> **Author Response (part 2 of 3)**
>
> ### **` Q3. Zero-shot performance of generative classifiers`**
> Thanks for the valuable insights. We evaluate two MLLMs zero-shot, Qwen2.5-VL-7B and VideoLLM-online-8B [3], on the COIN dataset (~750 actions). Qwen2.5 performs best, with 16.1% on step recognition and 8.9% on next action prediction; this significantly trails our finetuned GAD model (67.3% and 51.6%, see table B).  These results suggest that action understanding with large, semantically similar action vocabulary is still challenging for MLLMs, making fine-tuning essential.
>
> - Implementation: For Videollm-Online model, a procedural video understanding MLLM, we use it for open-ended generation. For Qwen2.5-VL, action prediction is treated either as an open-ended generation task, or by providing action labels as candidate options in the prompt. Post-processing is always applied to match the generation to action categories using the CLIP text encoder.
> - Extra Results: Providing action categories in the prompt worsens the performance, suggesting that including a large candidate set in the prompt can hinder MLLMs’ ability to correctly interpret instructions.
>
> [3] VideoLLM-online: Online Video Large Language Model for Streaming Video, CVPR, 2024
>
> *Table B. Zero-shot performance (accuracy) on existing MLLMs. Open-ended: free-form generation. "Categories in prompt": model selects from given categories. Post-processing maps outputs to action categories using CLIP text encoder and cosine similarity.*
>
> | Model                                | COIN Step (%) | COIN Next (%) |
> |--------------------------------------|---------------|---------------|
> | GAD (our finetuned model)            | 67.3          | 51.6          |
> | | | |
> | videollm-online-8B-v1plus            | 4.8          | 3.5           |
> | Qwen2.5-VL-7B (open-ended)           | 16.1          | 8.9           |
> | Qwen2.5-VL-7B (categories in prompt) | 11.9          | 6.5           |
>
>
> ### **` Q4. Generation-then-classification setting`**
> Thanks for bringing this point. We have the results showing that placing generation before classification degrades performance (Supplementary Table 9). Specifically,
> - Generation first.  During inference, the discriminative classifier tends to replicate the generative output, which is prone to confusion due to semantic overlap. This happens because conditioning on generation enables the discriminative classifier to take a shortcut, aligning its representation with the generation during training. Additionally, placing the learnable token at the end slows inference, as autoregressive generation must be completed first.
> - Classification first. Placing classification first enables the classifier to be less affected by generative results while still benefiting from semantic regularization and fast inference(skip generation).
> - We also ablate the parallel design, where discriminative and generative performance are less affected by interference. This simultaneous training resembles multi-task learning with a shared backbone, allowing the model to capture cross-task knowledge while benefiting from regularization. However, it requires two forward passes per output, reducing training efficiency. We will clarify this point in the main paper and please refer to additional results in Supp B.4.

---

> > ### Author Response · Authors · 2025-11-21
> > **Author Response (part 3 of 3)**
> >
> > ### **` Q5. CLS head scaling and inference with many categories`**
> > Thank you for your insightful observation. Based on our experiments in Table 1(main paper), scaling the classification head has little effect on inference speed and might improve performance.
> > - Inference speed. The majority of computation comes from the MLLM backbone rather than the language-modeling(LM) or classification head. For example, on COIN step recognition, the MLLM takes ~0.13 s per sample, while the LM head with 30k token vocabulary adds only ~0.0001 s (measured on an NVIDIA A40 GPU). Additionally, even with tens of thousands of categories (e.g., 20K), the computational cost remains small for classification: a single 20k-way linear layer is far cheaper than generating multiple tokens autoregressively over a 30k vocabulary.
> > -  Performance. Discriminative classifiers ignore semantic overlaps and establish clear decision boundaries. With more action categories, semantic overlaps rise, favoring discriminative learning. This can be reflected in our strong results on EPIC-Kitchens-100 (~3,600 actions) compared to other smaller datasets(main paper Table 1).
> >
> > ### **` Q6. Extra prior works`**
> > Thanks for pointing out these related prior works, we will add them in the revised paper.
> > Meanwhile, we would like to clarify their differences from our setting.
> > - Different from video QnA or MCQ tasks, action understanding faces a unique challenge: strong semantic overlap among action categories due to shared verbs and nouns. While those works mention a similar one-forward-pass strategy for efficiency,  our focus is on handling semantic overlaps in the text output space.
> > - Framing action understanding as an MCQ task is difficult in practice: listing many categories as options dilutes the instruction, and even performs worse than open-ended generation(e.g. Qwen2.5-VL model in Table B).

---

> ### Author Response · Authors · 2025-11-24
> **Clarification on faster training**
>
> Dear reviewer 4Sdb
>
> Thanks for your question, and sorry for the late reply.
>
> First, we would like to clarify that the reported training speedup refers to the comparison between discriminative and generative baselines. It is not a comparison between our GAD model and generative models. Since GAD also performs generation during training, this speedup does not apply to GAD itself. However, GAD is fast during inference because it can rely solely on the classification head and skip generation entirely (our design places the classification head before the generation module)
>
> 1. Length of training input.
> - For generative classifiers, the label text is tokenized into multiple tokens. For example, the label “add some oil to the pot” becomes {“add”, “some”, “oil”, “to”, “the”, “pot”}. Thus, the full input for training includes:
> > instruction tokens + video tokens + {“add”, “some”, “oil”, “to”, “the”, “pot”}.
>
> - For discriminative classifiers, we use only one learnable [CLS] token to query the class through a classification head. The input for training becomes simply:
> > instruction tokens + video tokens + {[CLS]}.
>
> 2. Gradient backpropagation.
> For generative classifiers, all label tokens are learnable. The loss is computed over the full sequence, e.g. {“add”, “some”, “oil”, “to”, “the”, “pot”}, which means gradients must backpropagate through each of these tokens. In contrast, a discriminative classifier only backpropagates through a single token, [CLS].
>
> 3. Classification head vs. language modeling head.
> It is true that the classification head and Language Modeling(LM) head differ in scale. However, our observations show that most of the computation comes from the MLLM backbone, not from the classification or LM head. For example, on COIN step recognition, the MLLM backbone takes approximately 0.13 s per sample, while an LM head with a 30k-token vocabulary adds only around 0.0001 s (measured on an NVIDIA A40 GPU).

---

> > ### Comment · Reviewer_4Sdb · 2025-11-24
> > **Concerns Resolved**
> >
> > Thank you to the authors for a comprehensive rebuttal clarifying all concerns. Rating updated to reflect this.
> >
> > Great work on an interesting paper!

---

> > > ### Author Response · Authors · 2025-11-24
> > >
> > > Dear reviewer 4Sdb
> > >
> > > Thank you very much for your valuable time and feedback and for raising your score. We are glad to have addressed your concerns. Your detailed and constructive comments have been very helpful, and we will revise the manuscript accordingly to further improve its clarity and quality.

---

### Official Review · Reviewer_fcN1 · 2025-10-31

**Soundness:** 3
**Presentation:** 4
**Contribution:** 3
**Rating:** 6
**Confidence:** 4

**Summary:**

The paper presents a study on generative vs discriminative methods for action classification, illustrating how the latter outperform the former in most cases. Several potential causes are mentioned in the paper, including the partitioning of actions into different tokens leading to overlapping and lack of distinctiveness in the representation of actions. The authors then propose a generative assisted classification approach where an LLM is endowed with a class token tasked with predicting actions in a closed-set environment, followed by a language decoder to decode past and future actions around the one classified with the CLS token. The paper is accompanied by supporting experiments illustrating the drawbacks of existing LLMs for generative action classification. Additional results show that their proposed approach can enhance existing LLMs for action classification.

**Strengths:**

The paper is very well written and presented. The motivation is clear and the paper is properly threaded. The authors identify specific challenges of generative models for action classification, and enumerate and study the possible reasons behind this subpar performance w.r.t. discriminative methods. Then, the authors devise a combined approach that is shown to produce better results. The paper is accompanied by proper ablation studies and an efficiency analysis, illustrating how discriminative (or single-token decoding) is not only more accurate in terms of closed-set prediction, but also faster at inference.

**Weaknesses:**

While the paper is well threaded and motivated, part of the story is driven towards obvious conclusions that leave aside some potential alternatives (please see the questions below). In particular,

1. The fact that discriminative approaches, or closed-set approaches, outperform generative ones, is not a finding or contribution of this paper, it is general knowledge. It is not expected that open-vocabulary models will outperform discriminative approaches in closed-set environments. The narrative is driven in a way that benefits discriminative models, without proper reference to the fact that this is constrained to the specific environments where the set of actions to be recognized is closed. In the questions below I suggest a potential study for a fairer comparison between generative and discriminative approaches. I also believe that bridging the gap between both methods is indeed making them the same (i.e. if a single token is used per action by extending the vocabulary, and the logits are capped to those of the target actions, then a generative approach boils down to a discriminative one with learnable queries and a one-hot binary classification approach).

2. It is a bit misleading that the narrative is built and tested in an ego-centric scenario where discriminative approaches are prone to thrive. How would the whole story sustain itself in a more generic scenario where several actors are performing potentially overlapping actions? E.g. the AVA setting where actions are not exclusive (i.e. in a multi-class scenario). In such case it looks to me that a generative approach can outperform discriminative approaches. If it is out of the scope of this paper to study such scenario, then it should be clearly stated in the paper to avoid leading to misinterpretations.

3. There's a missing text-to-text action retrieval from the predictions of generative approaches to closed-set predictions. In particular, it is not clear (or at least I missed that in the paper) how the generative predictions are mapped to actions.

4. There's no breakdown (unless I missed that, in which case I hope the authors can address this in the rebuttal) of where the confusing is coming from. In Fig. 3 the authors plot the false positives for the "add sugar" action, but it is not clear to me if this refers to action within the target set, or free-form actions.


In summary, I am borderline with the paper, as I believe it holds some valuable contribution, it is well written and presented, but lacks some proper contribution that is properly sustained.

**Questions:**

The runtimes in Table 1 illustrate that the discriminative counterparts of the corresponding LLMs are much faster than the generative ones. I understand that this might be due to the fact that when the actions are split into several tokens, it requires several forward calls to the model, incurring in slower inference. However, if the model is provided with action-specific tokenization as mentioned in l. 363, then inference should be achieved in a single forward pass. In such case the speed should be similar for both generative and discriminative approaches. In any case, what is the difference between a generative approach with extended vocabulary, whereby each full action corresponds to a token, and a discriminative approach with as many [CLS] learnable tokens as class labels? In the latter approach a one-vs-all discriminative approach could be learned (please refer to works on meta queries). I find that referring to the extended-vocabulary approach to generative is a bit overstretched. Can the authors elaborate on this?


In a closed-set setting with N classes, comparing discriminative and generative methods with the latter having a classifier that has L logits with L >> N is obviously biased. Have the authors tried to compare both methods by capping the classifier of the generative approach to N logits only, or to N' logits with N' being only the logits corresponding to the tokens involved in tokenizing the target classes?


How sensitive are the generative results to the tokenizer? I believe a proper study with further text encoders is necessary to study the influence of the tokenizers in the generative results.

How is the method trained in the case that there's a single action in a video? e.g. Kinetics?

---

> ### Author Response · Authors · 2025-11-21
> **Author Response (part 1 of 3)**
>
> Dear reviewer fcN1
>
> We greatly appreciate your thorough and constructive comments, and your recognition of the strength of our motivation and presentation, and more importantly the power of single-token decoding. We address the raised questions as follows.
>
> ### **`W1. The finding that discriminative outperforms generative models on close-set is general knowledge; generative and discriminative approaches converge by extending the vocabulary; a fair comparison between generative and discriminative models is needed.`**
>
> Thank you for bringing up these points to help us strengthen the paper. This “general knowledge” has yet to be explored in LLM-scale models for action understanding, which differs from most classification tasks as the labels often contain overlap semantics. In fact, state-of-the-art works [1,2,3] on action understanding rely exclusively on standard generative decoding. To our knowledge, we are the first to conduct a comprehensive analysis of discriminative approaches for video understanding tasks.
>
> Our findings also reveal how generative models can still be competitive, once the semantic overlaps in action labels are removed.
> - As you pointed out, extending actions to the tokenizer vocabulary (single token per action) should effectively eliminate their differences in both speed and performance, which aligns with our findings in Table 3 (main paper).
> - Mapping shared words among actions to distinct tokens can also close the performance gap (GAD_desync in Table 3). Here, the total number of tokens representing each action is unchanged.
> - For general video understanding tasks with distinct actions and fewer semantic overlaps, discriminative models will offer no clear advantage over generative ones.
>
> Moreover, we agree evaluating generative models on closed-set may be unfair compared to discriminative ones, and thank you for this point. In fact, our main intention is to understand  why they differ,  examine their respective strengths, and demonstrate how they can be effectively combined. We hope our findings on how semantic overlap affects generative models could encourage future work on designing better tokenization schemes that help generative models maintain strong discriminative ability. We also anticipate that this issue is not limited to the closed-set case and may similarly influence performance in open-set scenarios.
>
> [1] StreamMind: Unlocking Full Frame Rate Streaming Video Dialogue through Event-Gated Cognition, ICCV, 2025
>
> [2] VideoLLM-online: Online Video Large Language Model for Streaming Video, CVPR, 2024
>
> [3] VideoLLM-MoD: Efficient Video-Language Streaming with Mixture-of-Depths Vision Computation, NeurIPS 2024
>
>
> ### **`W2.  Ego-centric vs more general multi-actor, multi-class settings`**
> Thank you for your valuable suggestion. We evaluated our models on five different egocentric and exocentric video datasets, and our model is also suitable for multi-actor, multi-label scenarios. We will do our best to include these results before the rebuttal period ends. **[NOTE: we have now completed the experiments and shared the updated results in a separate thread for your reference.]**
>
> - Our experiments cover both egocentric videos (EPIC-Kitchens-100 and Ego4D GoalStep) and exocentric videos (THUMOS, CrossTask, and COIN).
> - Multi-label(actions are not exclusive) and single-label action understanding are usually treated as separate tasks. But for multi-label cases, we expect our claim on the Discriminative vs. Generative still holds as long as actions are semantically similar. The challenge lies in semantic overlap, rather than the multiple concurrent actions. Existing multi-label datasets rarely exhibit such semantic overlap, and for actions with clearer semantic separation (e.g., ‘run’ vs. ‘jump’), the advantage of discriminative over generative models is less obvious, even in single-label settings.
>
>
>
> ### **`W3. Mapping of generative predictions to closed-set actions`**
> Thanks for pointing this out. For fair comparison, we use Levenshtein edit distance [1, 2] for text-to-text action retrieval from the predictions of generative approaches to closed-set actions, though text-embedding-based matching is also possible. Since fine-tuning strongly favors memorization, over 99.5% of predictions exactly match the categories, making the choice of mapping method negligible.

---

> ### Author Response · Authors · 2025-11-21
> **Author Response (part 2 of 3)**
>
> ### **` W4. Origin of confusing false positives`**
> Thank you for highlighting this point. The origin of the confusion in generative models mainly comes from three categories: the first two are common factors that also affect discriminative classifiers, while the third is unique to generative models due to their text-based output.
> - The long-tail effect, where predictions are biased toward more frequent actions.
> - Visually similar actions, where actions look alike and produce similar input features.”
> - Semantic similar actions, where actions share vocabularies(e.g. verbs), causing the confusion in the output space.
>
> Our paper mainly focuses on the third confusion.
> - Fig. 3 in the paper shows the distribution of false positives after mapping the generative output to action categories, highlighting that generative models produce more diverse misclassifications than discriminative ones. For instance, the generative model may misclassify “add sugar” as “add spices” or “add meat” due to the shared verb “add,” whereas the discriminative classifier exhibits fewer semantically related errors. To clearly show this issue, we will provide additional explanation to Figure 3 and include further analysis of confusion matrix comparison in supplementary.
> - To quantify this diversity issue, we also introduce an entropy-based diversity score, which captures the variability of misclassified predictions; generative classifiers show higher misclassification diversity. Further details can be found in Supplementary B.3.
>
>
> ### **` Q1. Clarify and compare action-specific tokenization design and discriminative models`**
> Generative models with an extended vocabulary (each action as a single token) are conceptually similar to discriminative approaches (one [CLS] learnable token with a classification head), which is how we establish equivalence between the two types of models.
> - Inference speed. Extended vocabulary only requires one token generation, its inference speed should closely match that of a discriminative classifier. The majority of the computation lies in the LLM backbone, so even though the LM head has a much larger vocabulary (~30K tokens) than the classification head (typically 100–3,000 categories), it adds only a negligible overhead.
> - Performance. Table 3 (main paper) demonstrates comparable performance between the two approaches. Preserving each action label as a single token, rather than splitting it into shared words, resolves the semantic overlap issue and closes the performance gap.
> - Scalability. The extended vocabulary method is limited to specific scenarios. It requires that newly added categories never appear in the input query, since their token embeddings are initially unknown. But this method  should inspire designing new tokenizers that can support more conventional classification tasks as well as improve efficiency.
>
> Discriminative models can be adapted to a one-vs-all discriminative setting. We could either use multiple [CLS] learnable tokens, each representing a specific action, or a single [CLS] token with a sigmoid activation to predict multiple actions. Exploring this direction could be an interesting avenue for future work.  Thank you for raising this point.
>
> ### **` Q2. Capping generative logits to match discriminative setup`**
> Thank you for the valuable insights. Based on our empirical observation, capping generative logits to exclude unused tokens has no impact on the results.
> - Inference speed is dominated by the LLM backbone, so the presence of a heavier language head introduces only negligible overhead. For example, on COIN step recognition, the LLM takes ~0.13 s per sample, while the LM head adds only ~0.0001s (measured on an NVIDIA A40 GPU).
> - Unused tokens have no effect on performance:  they are never seen during training, so their logits constantly receive suppressive gradients, preventing them from being selected at inference time (over 99.5% predictions exactly match action categories). Since these tokens are not actively learned, keeping them in the vocabulary has no meaningful impact on the training dynamics or the final performance of the generative classifier.

---

> > ### Author Response · Authors · 2025-11-21
> > **Author Response (part 3 of 3)**
> >
> > ### **` Q3. Sensitivity of generative results to tokenizer choice and text encoders`**
> > Thank you for your careful consideration. We evaluated the tokenizer sensitivity on two widely used LLM families, Llama 3.2-1B and Qwen2.5-0.5B. Comparison results between discriminative and generative results are largely consistent across them (main paper Table 1).
> >
> > Besides, although the models differ in size and are not directly comparable, we observe that Llama generally outperforms Qwen2. This difference may stem from Llama’s larger hidden dimension (2048 vs. 896), which allows for more expressive representations(we always keep the vision encoder frozen).
> >
> >
> > ### **`Q4. Single-action video recognition task`**
> > Thank you for your valuable feedback. We include THUMOS in our evaluation, where most videos contain only a single action (results in Table 1 & 5 in the paper). We use their action labels for generation to capture semantics, improving performance over standard discriminative learning. However, when comparing discriminative to generative models, the improvement is less obvious given the simplicity of the dataset, highlighting its limited value in our setting.
> >
> > In addition, we focus on the impact of semantic overlap on generative models, especially in correlated (procedural videos) and compositional (verb–noun combinations) actions that reflect real-world complexity. Single-action video datasets like Kinetics, which typically include isolated and uncorrelated actions, are outside the scope of this study.

---

> ### Author Response · Authors · 2025-11-24
> **Author response (Updated results on general multi-actor, multi-label settings)**
>
> Dear reviewer fcN1
>
> We performed further experiments specifically addressing your inquiries about the multi-actor, multi-label (i.e., overlapping-action) scenario in Weakness 2. In this setting, we evaluate discriminative, generative classifiers and our GAD framework for multi-agent, multi-label action recognition.
>
> - **Dataset.** We evaluate on LEMMA dataset [1], which provides compositional atomic-action annotations in a multi-agent, multi-label environment. The dataset includes 739 unique atomic actions. The training split consists of 8,808 video segments, of which 553 contain multiple labels, while the test split has 2,886 segments, with 162 multi-label instances. Most multi-label segments involve two simultaneous actions, though some contain three.
> - **Implementation.** For the generative classifier, it is trained to generate the multi-label outputs sequentially. For the discriminative classifier, we use a single learnable [CLS] token and train with a binary cross-entropy loss. For our GAD, we follow the same classification strategy as the discriminative model, and use the action labels themselves as generation targets. During inference for both the discriminative classifier and GAD, we apply a sigmoid to the logits and predict all actions whose probabilities exceed a threshold, which we set to 0.3 in our experiments.
>
> - **Metrics.** We use two types of metrics for evaluation.
>   - Instance-wise accuracy. Each instance is treated as a single unit, regardless of whether it is single-label or multi-label. A prediction is considered correct only if it exactly matches the ground truth. For multi-label instances, any missing or extra predictions are counted as incorrect.
>   - Action-wise F1 score. Each action is considered separately. For each instance during testing, true positives (TP), false positives (FP), and false negatives (FN) are calculated action-wise. The final F1 score is then computed at the action level across all instances in the test set.  Here are two case examples for better understanding. Given an instance,
>     - Case a: Ground truth = {A, B}, Prediction = {B}  →   TP = 1, FP = 0, FN = 1
>     - Case b: Ground truth = {B}, Prediction = {A, B}  →  TP = 1, FP = 1, FN = 0
>
> - **Evaluation.** We report the performance of different models on all test segments (overall), as well as specifically on the subset of test segments with multiple labels (Multi-label) . The results indicate that discriminative classifiers consistently outperform generative ones on this dataset, even in the multi-label scenario. Furthermore, our GAD, which leverages semantics from the labels themselves, provides an additional regularization effect that further improves performance.
>
>
> *Table A. Performance comparison on multi-label action recognition task.*
> | Method         | Accuracy (Overall) | F1 Score (Overall) | Accuracy (Multi-label) | F1 Score (Multi-label) |
> |----------------|--------------------|---------------------|--------------------------|--------------------------|
> | Generative     | 44.9               | 45.5                | 4.3                      | 24.9                     |
> | Discriminative | 45.5               | 47.9                | 4.9                      | 30.7                     |
> | GAD            | 46.7               | 49.6                | 5.6                      | 32.1                     |
>
>
> [1] LEMMA: A Multi-view Dataset for LEarning Multi-agent Multi-task Activities, ECCV, 2020

---

### Official Review · Reviewer_CrzN · 2025-11-01

**Soundness:** 2
**Presentation:** 3
**Contribution:** 2
**Rating:** 4
**Confidence:** 3

**Summary:**

This paper investigates the use of Multimodal Large Language Models (MLLMs) for closed-set temporal action understanding. The authors identify a performance gap between generative and discriminative classifiers, attributing it to semantic overlap in label tokenization. To address this, they propose a Generation-Assisted Discriminative (GAD) classifier that incorporates an auxiliary generative objective during training to provide semantic regularization, while maintaining the discriminative model's efficient inference. Evaluations on multiple action understanding benchmarks show GAD achieves state-of-the-art results.

**Strengths:**

- The paper's greatest strength is its thorough, apples-to-apples comparison between generative and discriminative approaches across a wide range of tasks and datasets.
-  The implementation details in the main paper and appendix are extensive.

**Weaknesses:**

- Using a discriminative head on top of a generative model and training it with an auxiliary task is a standard practice in machine learning. The conceptual contribution is limited.
- The paper lacks a deep analysis of how the auxiliary generative task helps.
- The GAD framework is only evaluated in closed-set settings. Its applicability to open-world or few-shot scenarios is not discussed, limiting the scope of its claimed generality.

**Questions:**

- Given that the architecture of the discriminative classifier (appending a [CLS] token) and the idea of multi-task learning with an auxiliary objective are well-established, what is the specific conceptual novelty of the GAD framework beyond its application to the video modality?
- Can you design experiments to isolate the benefit of the auxiliary task? For instance, compare against a model trained with an equivalent amount of additional discriminative data or a different, non-generative auxiliary task to prove that the generative nature of the objective is key?

---

> ### Author Response · Authors · 2025-11-21
> **Author Response (part 1 of 2)**
>
> Dear reviewer CrZN
>
> We sincerely thank you for the time and valuable feedback. We are very pleased by your acknowledgement of our work's thorough comparison and extensive evaluation. Below we address your raised questions.
>
> ### **` W1. Limited conceptual novelty/contribution of the discriminative-head with auxiliary setup`**
> Our contribution lies not in the architectural choice itself, but in explaining why it is effective.  We provide the first systematic analysis of the trade-offs between generative vs discriminative classifiers in the MLLM setting, along with the tokenization strategies linking the two. These insights yield a deep characterization of MLLM-based action models and offer insights for future architectures that aim to combine generative semantics with discriminative accuracy.
> - Our analyses reveal a clear trade-off: generative classifiers capture rich semantics but suffer from label overlap and slower inference, while discriminative classifiers offer clean decision boundaries but limited semantic understanding.  Our architecture is a direct byproduct of these findings and could inspire future work on effectively leveraging the strengths of both approaches.
> - Our analyses show how standard tokenization harms generative behavior in tasks with label overlap, and we propose  improved tokenization strategies that help generative models retain strong discriminative ability.
> - Combining generative and discriminative modelling is non-trivial (ablations in Table. 9). Our GAD framework can jointly integrate a discriminative classifier for better action separation while incorporating a generative head as an auxiliary component to capture action semantics, serving as a strong, easily adaptable baseline for future research. Additionally, to our knowledge, GAD is the first MLLM-based framework for online action detection.
>
> ### **` W2. The benefit of the generative auxiliary task`**
> Thanks for bringing this point. Experimental results showing the benefits are demonstrated in Tables 4, 9, 11 (main paper) and Suppl. B.4.  We will emphasize these benefits in the revision.
> - Design advantages are shown in Table 4, comparing no generation, generating the action label itself, and generating contextual content such as previous actions. Generating context yields the best results. In addition, Table 9 presents an ablation study on the placement of the generative head, exploring configurations before, after, or in parallel with the classification head.  Placing it before classification is most effective, ensuring that classification is minimally affected by generated outputs while still benefiting generation regularization, and allows faster inference by skipping generation.
> - Benefits in capturing semantics: We compare against a non-generative auxiliary task: predicting the previous action as a classification task, similar to what the reviewer suggested. Table 11 shows that the generative nature of the auxiliary objective is crucial for performance, whereas using an auxiliary classification harms performance by disrupting the primary discriminative learning objective (see Suppl. B.4 for more results).
>
> In addition, thank you for the suggestion of a fair comparison against a model trained with additional discriminative data.
> To clarify, in our comparison, none of the models use any additional data. All methods rely solely on the provided annotations and are fine-tuned for the same number of epochs.

---

> > ### Author Response · Authors · 2025-11-21
> > **Author Response (part 2 of 2)**
> >
> > ### **` 3. Limited evaluation scope due to closed-set focus`**
> > Thank you for this comment. We appreciate the concern about open-set evaluation.
> >
> > If we understood correctly, the concern is about whether our approach restricts the underlying MLLM’s ability to operate in open-world or few-shot settings.
> > - To clarify, modern MLLMs are naturally strong in open-world scenarios due to their broad, data-driven pretraining; their weakness is instead in closed-set action understanding (see zero-shot performance in Table A), where specialized models still outperform them. Our work specifically targets this gap: we provide a simple mechanism that brings the MLLM up to state-of-the-art performance in closed-set action understanding.
> > - Regarding open-world applicability, the generative head is fully compatible with such settings, and we do not introduce any architectural changes that diminish the backbone MLLM’s generative capabilities. In fact, removing the LoRA adapter cleanly reverts the model to its standard, strong open-world behaviour.
> >
> > *Table A. Zero-shot performance (accuracy) on existing MLLMs. Open-ended: free-form generation. "Categories in prompt": model selects from given categories. Post-processing maps outputs to action categories using CLIP text encoder and cosine similarity.*
> >
> > | Model                                | COIN Step (%) | COIN Next (%) |
> > |--------------------------------------|---------------|---------------|
> > | GAD (our finetuned model)            | 67.3          | 51.6          |
> > | | | |
> > | videollm-online-8B-v1plus (open-ended)            | 4.8          | 3.5           |
> > | Qwen2.5-VL-7B (open-ended)           | 16.1          | 8.9           |
> > | Qwen2.5-VL-7B (categories in prompt) | 11.9          | 6.5           |
> >
> > As for our GAD model, because it is fine-tuned for a specific task, adapting it for action understanding inevitably reduces its generalization ability in open-world or few-shot settings. This behavior is a form of task-induced forgetting and is consistent with effects documented in the literature on other tasks [1,2].  Table B shows that predictions on the unseen Breakfast dataset increasingly collapse into training COIN categories, and the zero-shot performance decreases with training.  Balancing task-specific fine-tuning with open-world capability preservation is interesting, but as we mainly focus on action understanding, we leave this to future work.
> >
> > *Table B. Performance-generalization trade-off on our generative baseline. The model is trained on COIN task recognition and evaluated on both COIN and unseen Breakfast. Memorization rate is defined as the proportion of generated outputs matching training action categories during inference.*
> > | Epoch (iteration)           | Memorization Ratio (Breakfast)               |  COIN Accuracy (test set)      | Breakfast Accuracy(unseen) |
> > |------------------------|-----------------------------------|-------------------------------------|-----------------------------------|
> > | 0 (0)                    | 0.0                  | 8.28               | 6.33                              |
> > | 1 (521)                | 98.1                 | 77.0               | 4.45                              |
> > | 2 (1042)               | 99.6                 | 85.3               | 3.56                              |
> > | 3 (1563)               | 99.8                 | 91.3               | 3.59                              |
> > | 4 (2084)               | 99.9                 | 92.8               | 3.63                              |
> > | 5 (2600)               | 99.9                 | 92.7               | 3.62                              |
> >
> >
> > [1] Towards Calibrated Robust Fine-Tuning of Vision-Language Models, NeurIPS, 2024
> >
> > [2] Anchor-based Robust Finetuning of Vision-Language Models, CVPR, 2024
> >
> > [3] VideoLLM-online: Online Video Large Language Model for Streaming Video, CVPR, 2024

---

### Author Response · Authors · 2025-11-21
**Revision Note**

Dear All Reviewers :

We sincerely thank all the reviewers for their time and constructive feedback. We are pleased by the reviewers' positive acknowledgment of our thorough and extensive evaluation (Reviewers CrzN, fcN1, 4Sdb), clear presentation (Reviewers fcN1, 4Sdb),  good motivation and analysis (Reviewers fcN1), and interesting dual training strategy (Reviewers 4Sdb).

We have addressed the reviewers’ questions and concerns thoroughly in our individual responses. Please refer to the respective threads for details. We welcome any further feedback and would be glad to discuss and address it. Once again, we are truly grateful for your valuable efforts.

In addition, we have revised the paper to clarify ambiguous content and integrate new updates based on the reviewers’ feedback (highlighted in colored text). The key changes are:
- Add new related works on efficiently adapting generative models for multiple-choice question answering. in Sec. 2.
- Clarify our research focus on the fine-tuning stage and the rationale behind it in Sec. 3.
- Add explanation about how generative outputs are mapped to closed-set action labels in Sec. 4.1.
- Clarify the description of Fig. 3, and provide additional analysis in Suppl. E.
- Add explanation about why training speed is improved when using the discriminative classifier in Sec 4.2.
- Add ablation studies in Sec 4.3.
- Add the discussion of generalization trade-off in Sec 4.6.
- Expand discussion of limitations on generalization and corresponding experiments in Supple. C & D.

Thank you,

Authors

---

### Author Response · Authors · 2025-12-02
**Author Summary**

Dear reviewers, ACs,  SACs, and PCs,

Thank you for your time and effect in reviewing our submission. To assist the committee's evaluation, we would like to provide a concise overview of the reviewers’ feedback and our corresponding responses, along with the current rebuttal status.

- **Paper Strength.**  We are pleased by the reviewers' positive acknowledgment of our thorough and extensive evaluation (Reviewers CrzN, fcN1, 4Sdb), clear presentation (Reviewers fcN1, 4Sdb), strong motivation and analysis (Reviewers fcN1), and interesting methodology (Reviewers 4Sdb). Our work offers a timely analysis and design study of MLLM variants for action, with the key contributions summarized as follows:
  - As noted by Reviewer CrzN and fcN1, we conduct a rigorous analysis of generative and discriminative MLLM classifiers, discovering the key factors that distinguish their behavior  and highlighting their respective strengths: generative classifiers provide richer semantics but struggle with semantic overlap and slower inference, while discriminative models offer cleaner decision boundaries but weaker semantic coverage.
  - We demonstrate that standard tokenization harms generative behavior in tasks with semantic label overlap, and propose strategies that can help generative models retain strong discriminative capability, as acknowledged by Reviewer fcN1.
  - Building on these insights, we integrate both behaviors in our generative-assistant discriminative (GAD) framework, yielding a strong and easily adaptable baseline for future work, as recognized by Reviewer fcN1 and 4Sdb.

- **Paper Weakness and Our Response.** Most reviewers’ questions involved clarifications or experiments already included in supplementary. We have revised the paper to better emphasize these points. Beyond these, the main concerns centered on conceptual novelty, the functional role of the generative head in the proposed GAD framework, and clarification of generalization behavior.
  - *Novelty.* Reviewer CrzN questioned the simplicity of our architecture, but our key novelty lies in the analysis and findings of generative vs. discriminative classifiers, a key aspect recognized by other reviewers. Supplementary results further show that combining generative and discriminative components in our framework is non-trivial.
  - *Role of the generative head (Reviewers CrzN, 4Sdb).* Supplementary ablation studies examine the design and the function of our auxiliary generative head, showing where and how to use it to capture semantic richness to improve discriminative accuracy.
  - *Generalization behavior clarification (Reviewers CrzN, 4Sdb).* Our classification setting involves task-specific fine-tuning, which boosts action-classification performance but reduces general VLM capabilities, reflecting task-induced forgetting well-documented in the literature. We further provide experiments on a new dataset to evaluate generalization and show this trade-off. Importantly, the backbone architecture remains unchanged, and removing the introduced LoRA adapters can fully restore the model’s open-world performance.
  - *Extra experiments (Reviewers fcN1, 4Sdb).* We evaluate the zero-shot performance of existing MLLMs on action understanding to highlight the need for fine-tuning, and test performance in single-action and multi-label video settings.

- **Rebuttal status.**
   - On Nov. 24, Reviewer 4sDF confirmed that our clarifications resolved all concerns, raising the score from 6 to 8 and praising the paper as interesting and great work.
  - While the other two reviewers have not yet responded, Reviewer fcN1 initially gave a positive score (score 6), and we addressed all concerns with detailed experiments. Reviewer CrzN gave a low-confidence score, and we believe our clarifications adequately address his/her doubts.
  - We have revised the submission based on the reviewers’ feedback, with changes highlighted in colored text.

With the reviewers’ positive feedback and our rebuttal clarifications,  we believe the paper makes a meaningful contribution with potential impact. We kindly hope that the committee can take a thorough consideration of our paper’s contribution, and sincerely appreciate your time and consideration.

Best regards

Authors

---

### Meta-Review · Program_Chairs · 2026-01-13

**Summary:**

The paper attempts to reconcile discriminative and generative classifiers for action classification.
The reviewers' concerns revolved around the novelty of the finding (namely, that discriminative classifiers are better than generative ones in closed set action classification), the focus on closed set vs open set and multi-label scenarios, as well as questions about the specifics of the experimental protocols.

**Reviewer Concerns:**

I believe the concern about the novelty of the finding is not adequately addressed, though this is a subjective concern. The concern about the focus on closed set classification, vs open-set and multilabel scenarios is partially addressed: the authors provided additional experiments on a multilabel scenario. These experiments are well noted. The authors have also diligently answered all questions about the specifics of the method.

**Reviewer Scores:**

Reviewer 4Sdb explicitly stated that their concerns were resolved, and would have upgraded their rating to weak accept.

Based on the author response, reviewer fcN1 would also potentially have upgraded their rating given the additional experiments on multilabel scenarios.

Reviewer CrzN would likely have maintained their rating since the concern was the novelty of the finding.

---

### Decision · Program_Chairs · 2026-01-26

Accept (Poster)